# Tracking the neurodevelopmental trajectory of beta band oscillations with optically pumped magnetometer-based magnetoencephalography

Lukas Rier[1]*[†], Natalie Rhodes[1,2][†], Daisie O Pakenham[3], Elena Boto[1,4], Niall Holmes[1,4], Ryan M Hill[1,4], Gonzalo Reina Rivero[1], Vishal Shah[5], Cody Doyle[5], James Osborne[5], Richard W Bowtell[1], Margot Taylor[2], Matthew J Brookes[1,4]

[1]Sir Peter Mansfield Imaging Centre, School of Physics and Astronomy, University of Nottingham, University Park, Nottingham, United Kingdom; [2]Diagnostic Imaging, The Hospital for Sick Children, Toronto, Canada; [3]Clinical Neurophysiology, Nottingham University Hospitals NHS Trust, Queens Medical Centre, Nottingham, United Kingdom; [4]Cerca Magnetics Limited, 7-8 Castlebridge Office Village, Kirtley Drive, Nottingham, United Kingdom; [5]QuSpin Inc, Louisville, United States

*For correspondence:
lukas.rier@nottingham.ac.uk

[†]These authors contributed equally to this work

**Abstract** Neural oscillations mediate the coordination of activity within and between brain networks, supporting cognition and behaviour. How these processes develop throughout childhood is not only an important neuroscientific question but could also shed light on the mechanisms underlying neurological and psychiatric disorders. However, measuring the neurodevelopmental trajectory of oscillations has been hampered by confounds from instrumentation. In this paper, we investigate the suitability of a disruptive new imaging platform – optically pumped magnetometer-based magnetoencephalography (OPM-MEG) – to study oscillations during brain development. We show how a unique 192-channel OPM-MEG device, which is adaptable to head size and robust to participant movement, can be used to collect high-fidelity electrophysiological data in individuals aged between 2 and 34 years. Data were collected during a somatosensory task, and we measured both stimulus-induced modulation of beta oscillations in sensory cortex, and whole-brain connectivity, showing that both modulate significantly with age. Moreover, we show that pan-spectral bursts of electrophysiological activity drive task-induced beta modulation, and that their probability of occurrence and spectral content change with age. Our results offer new insights into the developmental trajectory of beta oscillations and provide clear evidence that OPM-MEG is an ideal platform for studying electrophysiology in neurodevelopment.

## eLife assessment

This study provides **important** evidence supporting the ability of a new type of neuroimaging, OPM-MEG system, to measure beta-band oscillation in sensorimotor tasks in 2-14 years old children and to demonstrate the corresponding development changes, since neuroimaging methods with high spatiotemporal resolution that could be used on small children are quite limited. The evidence supporting the conclusion is **compelling**. This work will be of interest to the neuroimaging and developmental science communities.

## Introduction

Neural oscillations are a fundamental component of brain function. They enable coordination of electrophysiological activity within and between neural assemblies and this underpins cognition and behaviour. Oscillations in the beta range (13–30 Hz) are typically associated with sensorimotor processes (*Barone and Rossiter, 2021*); they are prominent over the sensorimotor cortices, diminish in amplitude during sensory stimulation or motor execution (termed event-related decrease), and increase in amplitude (above a baseline level) following stimulus cessation (this is most often termed the post-movement beta rebound [PMBR] [*Pfurtscheller and Lopes da Silva, 1999*] in relation to movement). Beta oscillations and their modulation by tasks are robustly measured phenomena and their critical importance is highlighted by studies showing abnormalities across a range of disorders – e.g., autism (*Ronconi et al., 2020*), multiple sclerosis (*Barratt et al., 2017*), Parkinson's disease (*Little and Brown, 2014*), and Schizophrenia (*Gascoyne et al., 2021*). Despite this, little is known about the mechanistic role of beta oscillations, and most of what is known comes from studies applying non-invasive neuroimaging techniques to adult populations. Whilst the sensorimotor system changes little in adulthood, there are marked changes in childhood and a complete characterisation of the neuro-developmental trajectory of beta oscillations, particularly how they underpin behavioural milestones, might offer a new understanding of their role in healthy and abnormal function.

Several studies have investigated how neural oscillations change with age: *Gaetz et al., 2010*, measured beta modulation during index finger movement, showing that the PMBR was diminished in children compared to adults. *Kurz et al., 2016*, reported a similar effect when studying 11–19 year olds executing lower limb movement. *Trevarrow et al., 2019*, found an age-related increase in the PMBR amplitude in healthy 9–15 year olds, and further that the decrease in beta power during movement execution did not modulate with age. *Vakhtin et al., 2015*, showed an increase in PMBR amplitude between adolescence and adulthood, and that this trajectory was abnormal in autism. All these studies probed beta responses to movement execution; in the case of tactile stimulation (i.e. sensory stimulation without movement) both task-induced beta power loss and the post-stimulus rebound have been consistently observed in adults (*Pfurtscheller and Lopes da Silva, 1999*; *Gaetz and Cheyne, 2006*; *Cheyne et al., 2003*; *van Ede et al., 2010*; *Salenius et al., 1997*; *Cheyne, 2013*; *Kilavik et al., 2013*). Further, beta amplitude in sensory cortex has been related to attentional processes (*Bauer et al., 2006*) and is broadly thought to carry top-down influence on primary areas (*Barone and Rossiter, 2021*). However, there is less literature on how beta modulation changes with age during purely sensory tasks. A separate body of work has assessed neural oscillations in the absence of a task, demonstrating that there is a redistribution of oscillatory power across frequency bands as the brain matures. Specifically, low-frequency activity tends to decrease, and high-frequency activity increases with age (*Candelaria-Cook et al., 2022*; *Clarke et al., 2001*; *Whitford et al., 2007*). These changes are spatially specific, with increasing beta power most prominent in posterior parietal and occipital regions (*Hunt et al., 2019*; *Ott et al., 2021*). Beta oscillations are also implicated in long-range connectivity (*Brookes et al., 2011b*; *Brookes et al., 2011a*) and previous studies have demonstrated increased connectivity strength with age (*Schäfer et al., 2014*), particularly in attentional networks (*Brookes et al., 2018*). In sum, there is accord between studies that show increases in task-induced beta modulation and connectivity as well as a redistribution of spectral power, with increasing age.

Despite this progress, neurodevelopmental studies remain hindered by instrumental limitations. Neural oscillations can be measured non-invasively by either magnetoencephalography (MEG) or electroencephalography (EEG). MEG detects magnetic fields generated by neural currents, providing assessment of electrical activity with good spatial and millisecond temporal precision. However, the sensors traditionally used for field detection operate at low temperature, necessitating the use of fixed 'one-size-fits-all' sensor arrays. Because the signal declines with the square of distance, smaller head size leads to a reduction in signal (*Vorperian et al., 2007*). In addition, movement relative to fixed sensors degrades data quality. These limitations mean scanning young children with traditional MEG systems/SQUIDs is challenging and this has meant that most MEG studies on neurodevelopment are limited to older children and adolescents. Similarly, there are challenges in EEG. EEG measures differences in electrical potential across the scalp. The electrode array adapts to head shape and moves with the head, making it 'wearable' and consequently usable from new-borns to adults. However, the resistive properties of the skull distort signal topography, limiting spatial resolution.

Moreover, this confound changes with age as the skull increases in thickness (*Tröndle et al., 2022*). EEG is also more susceptible to interference from muscles than MEG (*Whitham et al., 2007*), particularly during movement. In sum, both EEG and MEG are limited; MEG is confounded by head size, EEG has poor spatial accuracy, and both are degraded by movement. However, in recent years, novel magnetic field sensors – optically pumped magnetometers (OPMs) – have inspired a new generation of MEG system (*Brookes et al., 2022*). OPMs are small, lightweight and have similar sensitivity to conventional MEG sensors but do not require cryogenics. This enables construction of a wearable MEG system (*Boto et al., 2018*). Because sensors can be placed flexibly, the array can adapt to head size and provide good coverage regardless of age. Further, because sensors move with the head, movement is possible during a scan. OPM-MEG is, ostensibly, ideal for children; e.g., Hill et al. showed the viability of OPM-MEG in a 2 year old (*Hill et al., 2019*); Feys et al. showed advantages for epileptic spike detection in children (*Feys et al., 2022*), and Corvilain et al. demonstrated utility even in the first weeks of life (*Corvilain et al., 2023*). However, no studies have yet used OPM-MEG in large groups to measure neurodevelopment.

In addition to instrumental limitations, most neurodevelopmental studies have used an approach to data analysis where signals are averaged over trials. This has led to the idea that sensory-induced beta modulation comprises a drop in oscillatory amplitude during movement and a smooth increase on movement cessation. However, recent studies (*Jones, 2016*; *Sherman et al., 2016*; *Shin et al., 2017*) investigating unaveraged signals show that, rather than a smooth oscillation, the beta rhythm is, in part, driven by discrete punctate events, known as 'bursts'. Bursts occur with a characteristic probability, which is altered by a task (*Little et al., 2019*; *Seedat et al., 2020*), and are not confined to the beta band but are pan-spectral, with components falling across many frequencies (*Gascoyne et al., 2021*; *Seedat et al., 2020*). There is also evidence that functional connectivity is driven by bursts that are coincident in time across spatially separate regions (*Seedat et al., 2020*). Recent work using EEG has found that, even in children as young as 12 months, beta band activity is driven by bursts (*Rayson et al., 2022*). Further work, also using EEG, investigated burst activity in infants (9 and 12 months)

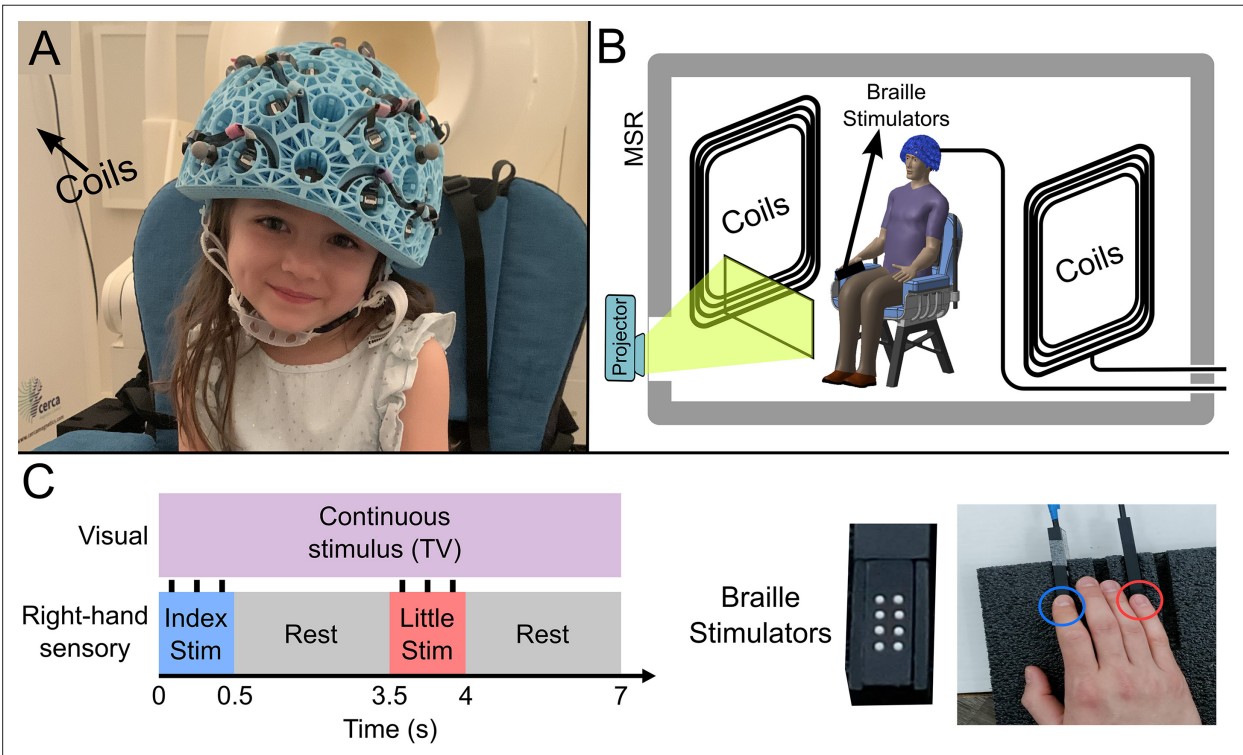

**Figure 1.** Experimental setup and beta band modulation during sensory task. (**A**) 4-year-old child wearing an optically pumped magnetometer-based magnetoencephalography (OPM-MEG) helmet (consent and authorisation for publication was obtained). (**B**) Schematic diagram of the whole system inside the shielded room. (**C**) Schematic illustration of stimulus timings and a photo of the somatosensory stimulators. 'Braille' stimulators each comprise eight pins, which can be controlled independently; all eight were used simultaneously to deliver the stimuli.

and adults during observed movement execution, with results showing stimulus-induced decrease in burst rate at all ages, with the largest effects in adults (*Rayson et al., 2023*). These studies have changed the way that the research community thinks about beta oscillations (*van Ede et al., 2018*) and a full understanding of beta dynamics and their age dependence must be placed in the context of the burst model.

Here, we combine OPM-MEG with a burst analysis based on a hidden Markov model (HMM) (*Seedat et al., 2020*; *Baker et al., 2014*; *Vidaurre et al., 2016*) to investigate beta dynamics. We aimed to scan a cohort of children and adults across a wide age range (upwards from 2 years of age). Because of this, we implemented a passive somatosensory task which can be completed by anyone, regardless of age. Our study addresses two objectives: First, we test the veracity of a novel 192-channel triaxial OPM-MEG system for use in paediatric populations, investigating its practicality in young children and assessing whether previously observed age-related changes in task-induced beta modulation and functional connectivity can be reliably measured using OPM-MEG. Second, we investigate how task-induced beta modulation in the sensorimotor cortices is related to the occurrence of pan-spectral bursts, and how the characteristics of those bursts change with age.

## Results

Our OPM-MEG system comprised a maximum of 64 OPMs (QuSpin Inc, Colorado, USA), each capable of measuring magnetic field independently in three orthogonal orientations, meaning data were recorded using up to 192 channels. Sensors were mounted in 3D-printed helmets of differing size (Cerca Magnetics Ltd. Nottingham, UK), allowing adaptation to the participant's head (*Figure 1A*). The total weight of the helmet ranged from ~856 g (in the smallest case) to ~906 g (in the largest case). The system was integrated into a magnetically shielded room (MSR) equipped with an active field control system (see 'coils' in *Figure 1A and B*; Cerca Magnetics Ltd. Nottingham, UK) which allowed reduction of background field to <1 nT. This was to ensure that participants were able to move during a scan without compromising sensor operation (*Borna et al., 2017*; *Holmes et al., 2018*). A schematic of the system is shown in *Figure 1B*.

27 children (aged 2–13 years, 17 female) and 26 adults (aged 21–34 years, 13 female) took part in the study. All participants performed a task in which two stimulators (*Figure 1C*) delivered passive somatosensory stimulation to either the index or little finger of the right hand sequentially. Stimuli lasted 0.5 s, occurred every 3.5 s, and comprised three taps on the fingertip. This pattern of stimulation

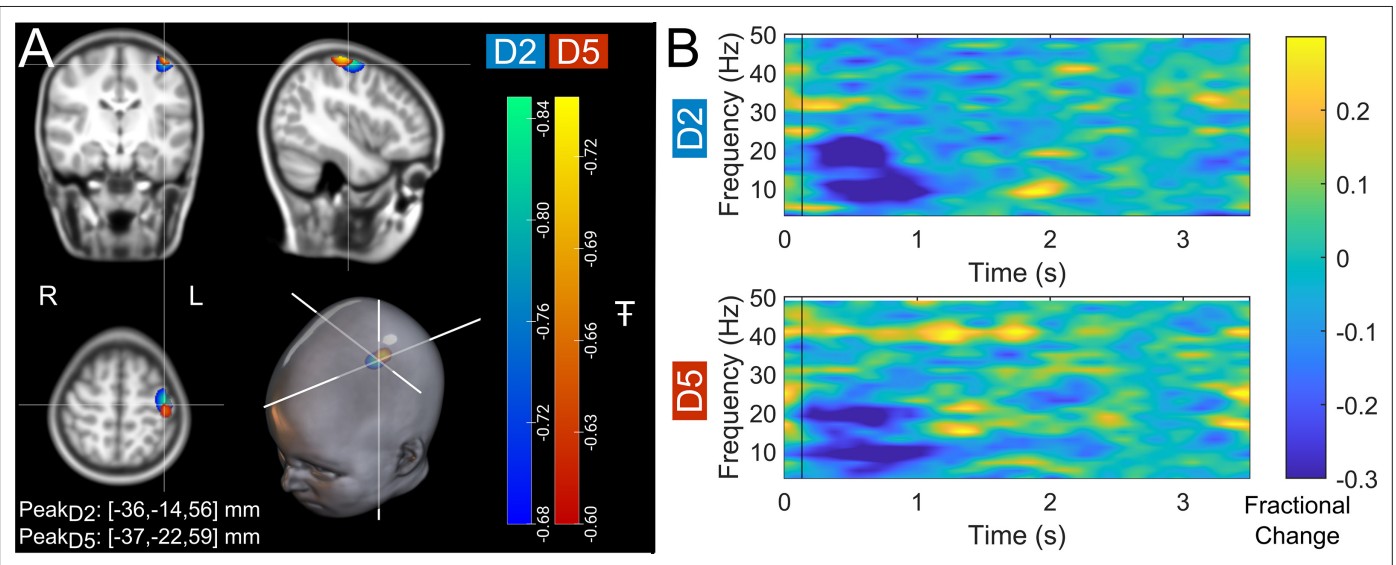

**Figure 2.** Data from a single participant (7 years of age). (**A**) Brain plots show slices through the left motor cortex, with a pseudo-T-statistical map of beta modulation. The blue/green peaks indicate locations of largest beta modulation during stimulation for index finger trials (digit 2/D2), while the red/yellow peaks show the little finger (digit 5/D5). (**B**) Time-frequency spectra showing neural oscillatory amplitude modulation (fractional change in spectral amplitude relative to baseline measured in the 2.5–3 s window) for both fingers, using data extracted from the location of peak beta modulation (left sensorimotor cortex). Vertical lines indicate the time of the first braille stimulus. Note the beta amplitude reduction during stimulation, as expected.

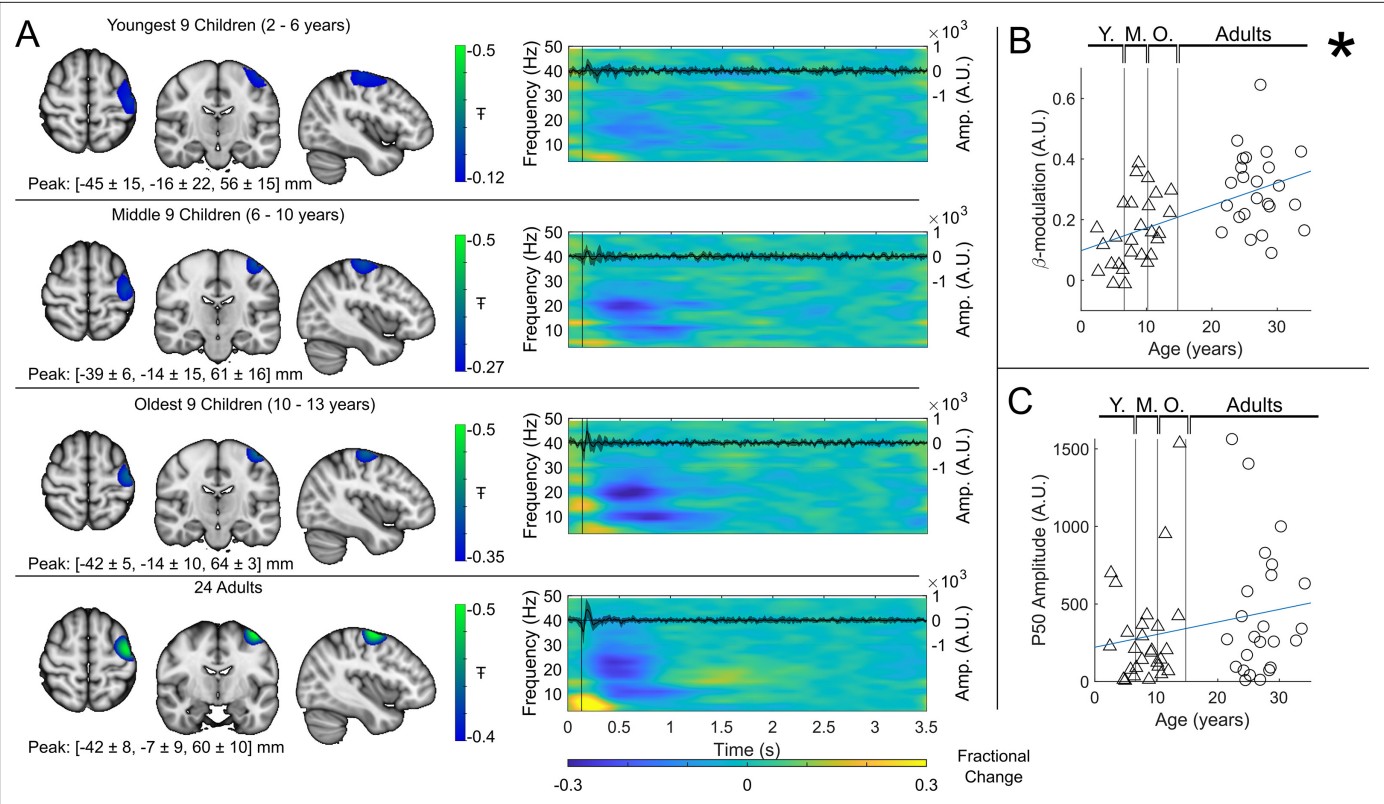

**Figure 3.** Beta band modulation with age (index finger). (**A**) Brain plots show slices through the left motor cortex, with a pseudo-T-statistical map of beta modulation (blue/green) overlaid on the standard brain. Peak MNI coordinates are indicated for each subgroup. Time-frequency spectrograms show modulation of the amplitude of neural oscillations (fractional change in spectral amplitude relative to the baseline measured in the 2.5–3 s window). Vertical lines indicate the time of the first braille stimulus. In all cases results were extracted from the location of peak beta desynchronisation (in the left sensorimotor cortex). Note the clear beta amplitude reduction during stimulation. The inset line plots show the 4–40 Hz trial averaged phase-locked evoked response, with the expected prominent deflections around 20 ms and 50 ms. Shaded areas indicate the standard deviation of the evoked traces across the group. (**B**) Maximum difference in beta band amplitude (0.3–0.8 s window vs 1–1.5 s window) plotted as a function of age (i.e. each data point shows a different participant; triangles represent children, circles represent adults). Note significant correlation ($R^2 = 0.29, \mathrm{p} = 0.00004^*$). (**C**) Amplitude of the P50 component of the evoked response plotted against age. There was no significant correlation ($R^2 = 0.04, \mathrm{p} = 0.14$). All data here relate to the index finger stimulation; similar results are available for the little finger stimulation in *Figure 3—figure supplement 1*.

The online version of this article includes the following figure supplement(s) for figure 3:

**Figure supplement 1.** Beta band modulation with age (little finger).

was repeated 42 times for both fingers. Throughout the experiment, participants could watch their favourite TV show. Following data preprocessing, high-fidelity data were available in 27 children and 24 adults. Two datasets were excluded from further analysis as data quality was not sufficient to perform our HMM analysis (see Methods). We removed 19±12% (mean ± standard deviation) of trials in children, and 9±5% of trials in adults due to excessive interference. On average we had 160±10 channels with high-quality data available (note that not all sensors were available for every measurement – see also Discussion).

## Beta band modulation with age

*Figure 2* shows beta band modulation during the task for a single representative child (7 years of age). Panel A shows the estimated brain anatomy (see Methods) with the locations of the largest beta modulation overlaid – contrasted between stimulus (0.3–0.8 s relative to stimulus onset) and rest (2.5–3 s) time windows. Data for index and little finger simulation are overlaid in blue/green and red/yellow, respectively. The largest effects fall in the sensorimotor cortices as expected. Panel B shows time-frequency spectra depicting the temporal evolution of the amplitude of neural oscillations. Blue represents a decrease in oscillatory amplitude relative to baseline (2.5–3 s); yellow represents an increase. As expected, there is a reduction in beta amplitude during stimulation.

Group averaged beta dynamics are shown in *Figure 3*. Here, for visualisation, the children were split into three groups of 9: youngest (aged 2–6 years), middle (6–10 years), and oldest (10–13 years). Data were averaged within each group, and across all 24 adults (21–34 years) for comparison. The brain plots show group averaged pseudo-T-statistical maps of stimulus-induced beta band modulation. In all groups, a modulation peak appeared in the left sensorimotor cortex. We observed no significant difference in the location of peak beta modulation between index and little finger stimulation (see also Discussion). The time-frequency spectrograms (TFSs) are also shown for each group. Here, we observe a drop in beta amplitude during stimulation for all three groups, however this was most pronounced in adults and was weaker in younger children. For statistical analysis, we estimated the maximum difference in beta band amplitude between stimulation (0.3–0.8 s) and post-stimulation (1–1.5 s) windows and plotted this as a function of age (*Figure 3B*). Here, Pearson correlation suggested a significant ($R^2 = 0.29, \mathrm{p} = 4 \times 10^{-5}$) relationship. These data agree strongly with previous studies showing increased task-induced beta modulation with age (though here we present a sensory, rather than motor task). However, they are acquired using a fundamentally new wearable technology, and in younger participants.

For completeness, the inset time course within each time-frequency plot shows the beamformer-projected trial and subject averaged evoked response in sensorimotor cortex (estimated by trial averaging the beamformer-projected data in the 4 Hz to 40 Hz band). Again, there is a neurodevelopmental effect with a significant increase in M50 amplitude with age in the little finger (see *Figure 3— figure supplement 1*, $R^2 = 0.1, \mathrm{p} = 0.023$) though this did not reach significance in the index finger (*Figure 3C*, $R^2 = 0.04, p = 0.14$).

## Functional connectivity in the beta band

Whole-brain beta band functional connectivity was estimated by calculating amplitude envelope correlation (AEC) (*O'Neill et al., 2015*) between (unaveraged) beta band signals extracted from 78 cortical regions. *Figure 4A* shows connectome matrices averaged across participants in each of the four groups; each matrix element represents the strength of a connection between two brain regions. In the 'glass brains', the red lines show the 150 connections with the highest connectivity. In adults, the connectome is in strong agreement with those from previous studies (*Schäfer et al., 2014*; *Boto et al., 2021*), with prominent sensorimotor, posterior-parietal- and fronto-parietal connections. However, connectivity patterns in children differed in both strength and spatial signature, with the visual network showing the strongest connectivity. To statistically test the relationship between connectivity and age, we plotted global connectivity (i.e. the sum of all matrix elements) versus age (*Figure 4B*). Pearson correlation suggested a significant ($R^2 = 0.42, \mathrm{p} = 2.67 \times 10^{-7}$) relationship with older participants having increased connectivity. We also probed how this relationship changes across brain regions: *Figure 4D* shows example scatter plots of node degree (i.e. how connected a specific region is to the rest of the brain) for two pairs of homologous frontal and occipital regions. Note that the gradient of the fit in the frontal regions (0.27 $\mathrm{year}^{-1}, R^2 = 0.44, \mathrm{p} = 1.2 \times 10^{-7}$ and 0.27 $\mathrm{year}^{-1}, R^2 = 0.50, \mathrm{p} = 5.8 \times 10^{-9}$) is much larger than that in the occipital regions (0.10 $\mathrm{year}^{-1}, R^2 = 0.18, \mathrm{p} = 2.0 \times 10^{-3}$, and 0.12 $\mathrm{year}^{-1}, R^2 = 0.29, \mathrm{p} = 4.2 \times 10^{-5}$). This is delineated for all brain regions in *Figure 4C*, where each region is coloured according to the gradient of the fit. The regions showing the largest change with age are frontal and parietal areas, with visual cortex demonstrating the smallest effect.

## Burst interpretation of beta dynamics

To assess pan-spectral bursts, we applied a univariate, three-state HMM to the broadband (1–48 Hz) electrophysiological signal extracted from the location of largest beta modulation. This enabled us to identify the times at which bursts occurred in sensorimotor cortex (*Seedat et al., 2020*; *Rier et al., 2021*).

*Figure 5A* shows a raster plot of burst occurrence for all individual task trials in all participants. White represents time points and trials where bursts are occurring; black represents the absence of a burst. Participants are separated by the red lines and groups are separated by the blue lines. Burst absence is more likely in the time period during stimulation, indicating a task-induced decrease in burst probability. *Figure 5B* shows group averaged burst probability as a function of time. In all age groups, bursts were less likely during stimulation, though this modulation changes with age, with the

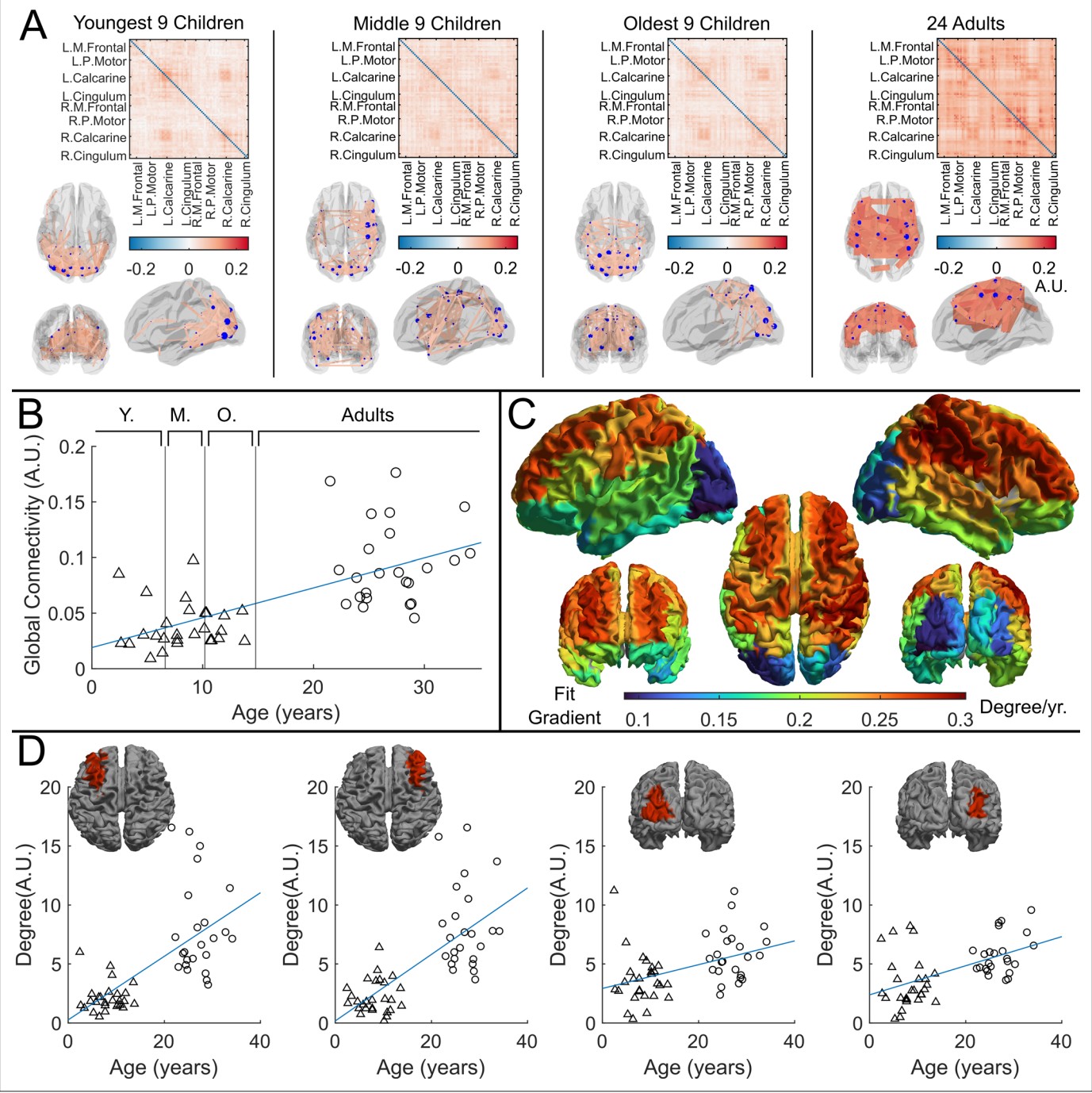

**Figure 4.** Functional connectivity – estimated using amplitude envelope correlation (AEC) – varies with age. (**A**) Connectivity matrices constructed using 78 regions of the automated anatomical labelling (AAL) atlas and glass brains showing the strongest 150 connections (average across the group). AEC was estimated across the entire recording. (**B**) Global average connectivity increases significantly with age ($R^2 = 0.42$, p = $2.67 \times 10^{-7}$*). (**C**) Age-related changes in connectivity vary spatially. Brain plot shows the linear fit gradient of node degree (the sum across the rows of the connectivity matrices) against age. Node degree varies less in occipital regions while frontal regions become more strongly connected with increasing age. (**D**) Example plots show node degree against age for left and right frontal and occipital regions. Pearson correlation yielded (from left to right): ($R^2 = 0.44$, p = $1.2 \times 10^{-7}$, Degree = $0.27 \cdot$ age + 0.26); ($R^2 = 0.50$, p = $5.8 \times 10^{-9}$, Degree = $0.28 \cdot$ age + 0.17); ($R^2 = 0.18$, p = $2.0 \times 10^{-3}$, Degree = $0.10 \cdot$ age + 2.92); ($R^2 = 0.29$, p = $4.2 \times 10^{-5}$, Degree = $0.12 \cdot$ age + 2.38).

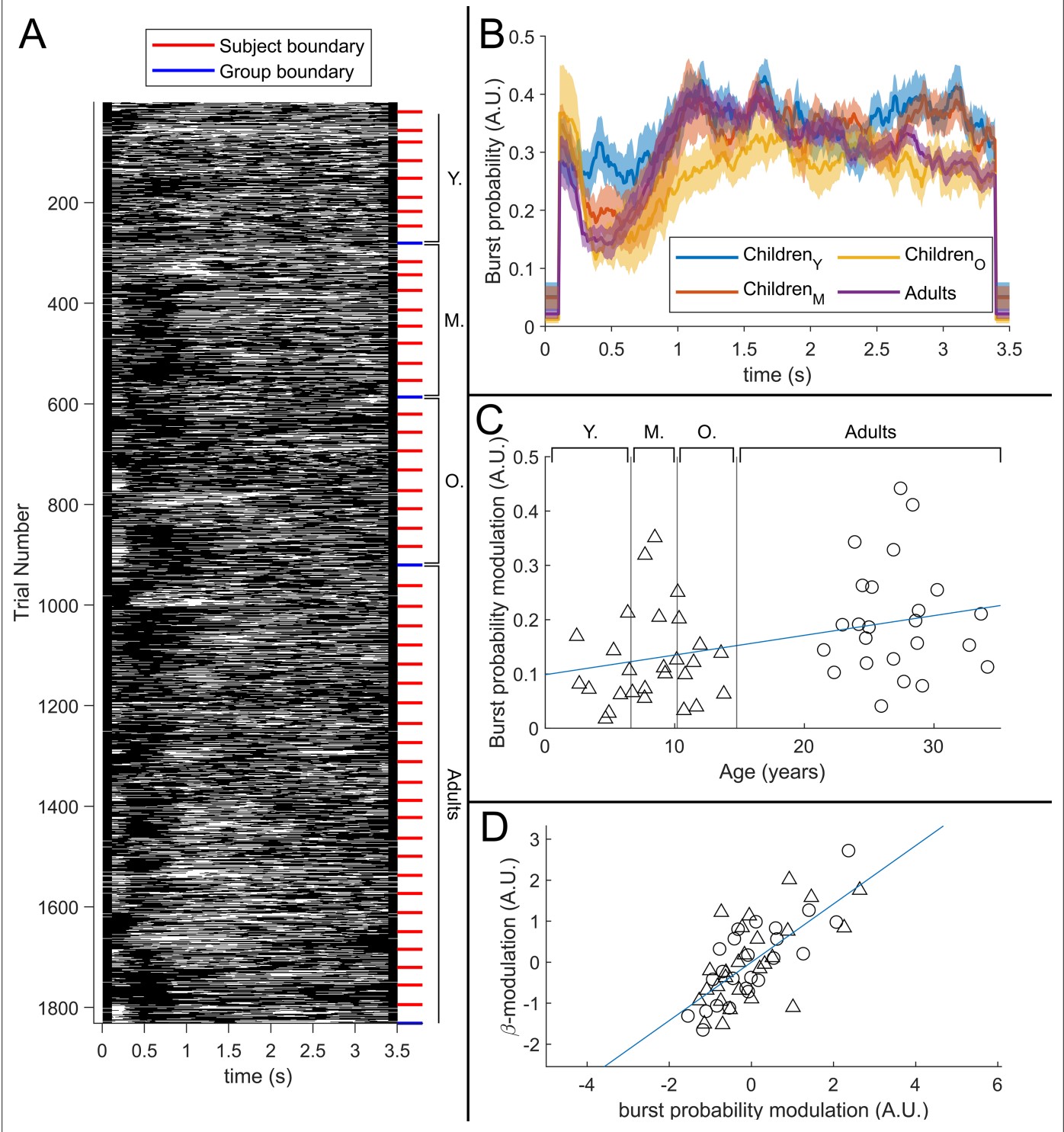

**Figure 5.** The relationship between beta band amplitude modulation and pan-spectral burst probability. (**A**) Raster plot showing burst occurrence (white) as a function of time for all trials and participants combined (participants sorted by increasing age). (**B**) Trial averaged burst probability time courses across the four participant groups. Shaded areas indicate the standard error. (**C**) Stimulus to post-stimulus modulation of burst probability (0.3–0.8 s vs 1–1.5 s) plotted against age. Note significant ($R^2 = 0.13$, $p = 0.0089*$) positive correlation. (**D**) Beta amplitude modulation plotted against burst probability. Note again significant correlation ($R^2 = 0.5$, $p = 5.2 \times 10^{-9}*$). Values for both measures were z-transformed within the children and adult group respectively to mitigate the age confound. Triangles and circles denote children and adults respectively.

younger group demonstrating the least pronounced effect. This is tested statistically in *Figure 5C* which shows a significant ($R^2 = 0.13, \mathrm{p} = 8.9 \times 10^{-3}$*) positive Pearson correlation between the modulation of burst probability and age. *Figure 5D* shows the association between beta amplitude and burst probability modulation. Here, the significant ($R^2 = 0.50, \mathrm{p} = 5.2 \times 10^{-9}$*) positive relationship supports a hypothesis that the observed change in task-induced beta modulation with age (shown in *Figure 3*) is driven by changes in the modulation of burst probability. Interestingly, we saw no measurable change in the amplitude of bursts with age (see Appendix 1).

We estimated the spectral content of the bursts identified by the HMM. In *Figure 6A* the burst spectra are shown for all four participant groups. In adults, the spectral power diminishes with increasing frequency, with additional peaks in the alpha and beta band. In children, high frequencies are diminished, and low frequencies are elevated, compared to adults. This is also shown in *Figure 6B* where, for every frequency, we perform a linear fit to a scatter plot of spectral density versus age. Here, positive values indicate that spectral power increases with age; negative power means it decreases. The inset scatter plots show example age relationships at 3 Hz, 9 Hz, 21 Hz, and 37 Hz. We see a clear decrease in low-frequency spectral content and increasing high-frequency content, with age. Interestingly, spectral content in the alpha band appeared stable with no significant correlation with age. Similar trends for changes in frequency content with age were found for the non-burst states (see *Figure 6—figure supplement 1*).

## Discussion

There are few practical, non-invasive platforms capable of measuring brain function in children. Functional magnetic resonance imaging (*Ogawa et al., 1990*) tracks brain activity with millimetre resolution, but the mechanism of detection is indirect (based on haemodynamics) with limited temporal precision. Participants must also lie immobile and are exposed to high acoustic noise; many children find this challenging and it is difficult to implement naturalistic experiments. Functional near infra-red spectroscopy (fNIRS) (*Chance et al., 1993*) provides a wearable platform which allows scanning of almost any participant during any conceivable experiment. However, fNIRS is also restricted to haemodynamic metrics; it has limited temporal resolution and spatial resolution is ~1 cm. EEG (*Berger, 1929*) measures electrophysiological activity in neural networks and thus offers millisecond temporal precision. It is also wearable, adaptable to any participant, and enables naturalistic experiments. However, spatial resolution is restricted due to the inhomogeneous conductivity profile of the head. This problem is exacerbated in young (<18 months) children due to additional inhomogeneities caused by the fontanelle, and in neurodevelopmental studies due to changing skull thickness. EEG is also highly susceptible to artefacts from electrical activity in muscles. Conventional MEG (*Hämäläinen et al., 1993*) measures brain electrophysiology with both high spatial and temporal resolution, but is limited in performance and practicality due to the fixed nature of the sensor array. It follows that the technologies currently in use for neurodevelopmental assessment are limited by either practicality, performance, or both. OPM-MEG ostensibly offers the performance of MEG, with the practicality of fNIRS/EEG, making it attractive for use in children. Here, our primary aim was to test the feasibility of OPM-MEG for neurodevelopmental studies. Our results demonstrate we were able to scan children down to age 2 years, measuring high-fidelity electrophysiological signals and characterising the neurodevelopmental trajectory of beta oscillations. The fact that we were able to complete this study demonstrates the advantage of OPM-MEG over conventional MEG, the latter being challenging to deploy across such a large age range.

### System design for neurodevelopmental studies

We designed our system for lifespan compliance. Multiple sizes of helmet allowed us to select the best fitting size for any given participant. A statistical analysis (see Appendix 2) showed no significant change in scalp-to-sensor distance with age, meaning sensors were not further away from the scalp in children (who tended to have a smaller head circumference) than they were in adults. Additional simulations suggested that, had our cohort been scanned in a single helmet size, sensor proximity would have been a confound. This is an important point which demonstrates the advantages of an adaptable OPM-MEG array over a static array. Relatedly, it is noteworthy that an analysis of beta burst amplitude showed no measurable modulation with subject age (Appendix 1); this (indirectly) suggests

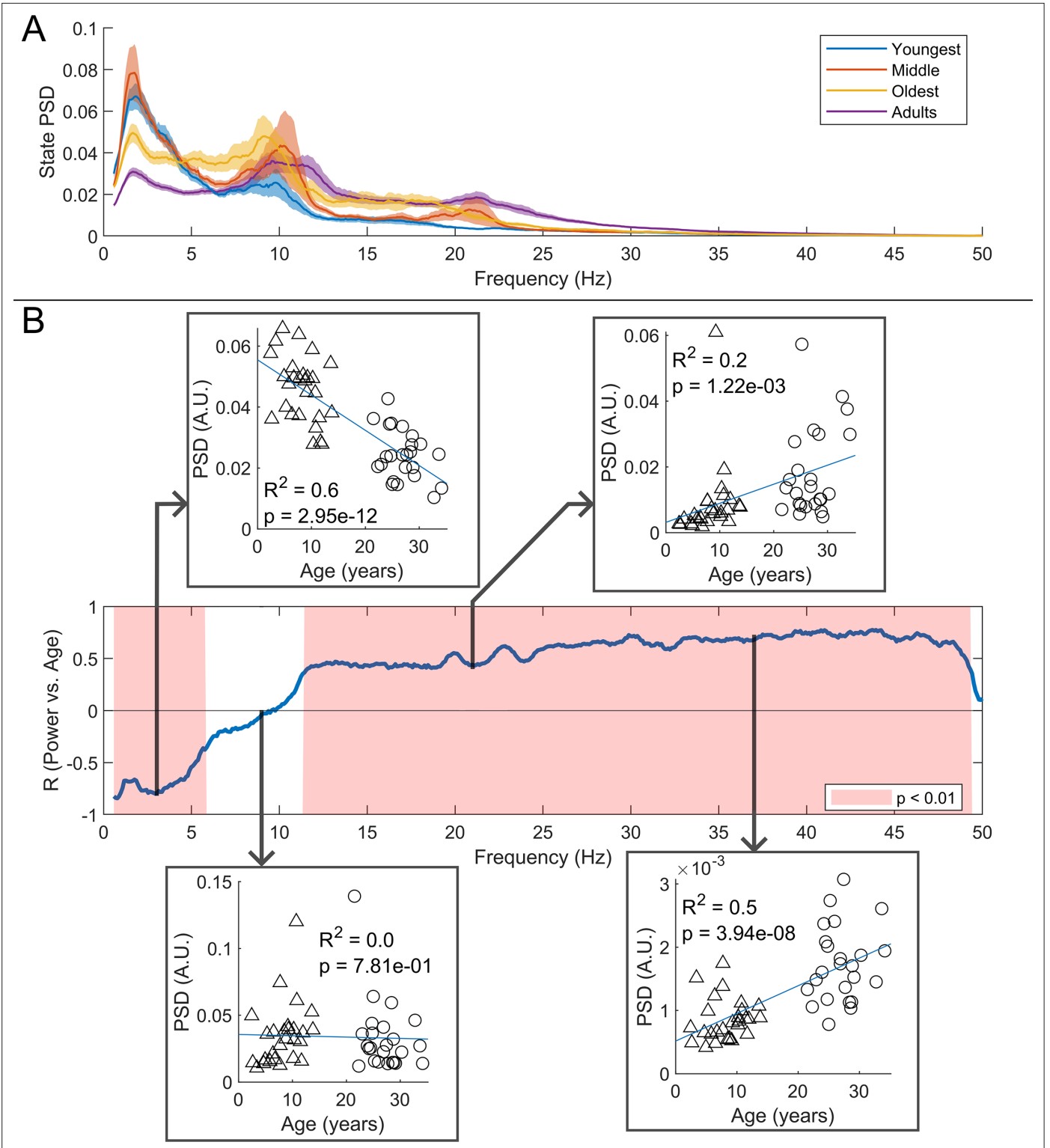

**Figure 6.** Spectral content of the burst state varies with age. (**A**) Average burst-state spectra across groups. Shaded areas indicate standard error on the group mean. (**B**) Pearson correlation coefficient for the power spectral density (PSD) values in (**A**) against age across all frequency values. Red shaded areas indicate $p < 0.01$ (uncorrected). The four inset plots show example scatters of PSD values with age at selected frequencies (3 Hz, 9 Hz, 21 Hz, and 37 Hz). Low-frequency spectral content decreases with age while high-frequency content increases. No significant correlation was observed in the high theta and alpha bands.

The online version of this article includes the following figure supplement(s) for figure 6:

**Figure supplement 1.** Spectral content of the non-burst states.

we are not losing sensitivity in the youngest volunteers (if we were this would presumably result in lower amplitude bursts in children). The helmets themselves were relatively lightweight, ranging from ~856 g (in the smallest case) to ~906 g (in the largest case). While this is heavier than, for example, a child's bicycle helmet (the average weight of which is ~300–350 g) they were well tolerated by our cohort. Heat from the sensors (which require elevated temperature to maintain operation in the spin exchange relaxation free regime *Allred et al., 2002*) was controlled via both convection (with air being able to flow through the helmet lattice) and an insulating material cap worn under the helmet by all participants (see *Figure 1A*). Together, these ensured that participants remained comfortable throughout data recording.

Whilst the helmet allows sensors to move with the head, sensor operation is perturbed by background fields (e.g. if a sensor rotates in a uniform background field, or translates in a field gradient, it will see a changing field which can obfuscate brain activity and, in some cases, stop the sensors working; *Boto et al., 2018*). For this reason, our system also employed active field control (*Holmes et al., 2018*) which enabled us to reduce the field to a level where sensors work reliably, even in the presence of head movements. This meant that, although we did not encourage our participants to move, they were completely unrestrained. The sensors themselves are also robust to head motion, as every sensor is a self-contained unit connected to its own control electronics by a cable that can accommodate rapid and uncontrolled movement. One limitation of the current study is that practical limitations prevented us from quantitatively tracking the extent to which children (and adults) moved their head during a scan. Anecdotally, however, experimenters present in the room during scans reported several instances where children moved, for example, to speak to their parents who were also in the room. Such levels of movement could not be tolerated in conventional MEG or MRI and so this again demonstrates the advantages afforded by OPM-MEG.

There were two other design features which helped ensure our system was optimal for children. Firstly, a challenge when imaging children is the proximity of the brain to the scalp; the brain-scalp separation is ~15 mm in adults but can be as little as ~5 mm in children. Previous work (*Boto et al., 2022*) has shown that, when using radially oriented magnetic field measurements, a combination of finite sampling and brain proximity leads to inhomogeneous coverage (i.e. spatial aliasing). Here, our system was designed with triaxial sensors which helps to prevent this confound (we also note that triaxial sensors enable improved noise rejection; *Brookes et al., 2021*; *Tierney et al., 2022*). Secondly, our system was housed in a large MSR which allowed children to be accompanied by a parent and experimenter throughout the scan. These features led to a system that enables acquisition of high-quality MEG data and is also well tolerated.

Ultimately, we obtained usable data in 27/27 children and 24/26 adults. Our findings support previous neurodevelopmental studies (*Gaetz et al., 2010*; *Kurz et al., 2016*; *Trevarrow et al., 2019*; *Schäfer et al., 2014*) and in this way validate OPM-MEG by showing substantial equivalence to the established state-of-the-art. Importantly, however, most prior studies of neurodevelopmental trajectory in MEG were carried out in older children – e.g., *Kurz et al., 2016*, showed a similar effect in 11–19 year olds; *Trevarrow et al., 2019*, employed a cohort of 9–15 year olds, and our own previous work also scanned a cohort of 9–15 year olds (*Brookes et al., 2018*). In the present study, we were able to successfully scan children from age 2 years and there are no fundamental reasons why we could not have scanned even younger participants. There are important reasons for moving to younger participants: from a neuroscientific viewpoint, many critical milestones in development occur in the first few years (even months) of life – such as learning to walk and talk. If we can use OPM-MEG technology to measure the brain activities that underpin these developmental milestones, this would offer a new understanding of brain function. Moreover, many disorders strike in the first years of life – e.g., autism can be diagnosed in children as young as 2 years and epilepsy has a high incidence in children, including in the neonatal and infant period (*Specchio et al., 2022*). In those where seizures cannot be controlled by drugs, surgery (which can be informed by MEG assessment) is often a viable option for treatment; the younger the patient, the more successful the outcome (*Lamberink et al., 2020*). For these reasons, the development of a platform that enables the assessment of brain electrophysiology, with high spatiotemporal precision, in young people is a significant step and one that has potential to impact multiple areas.

Although the system was successful, there are some limitations to the present design which should be mentioned. Firstly, the range of available helmets was limited, and future studies may aim to use

more sizes (or flexible helmets) to better accommodate variation in head size and shape. Also, even the lightweight helmet used here may be too heavy for younger participants; whilst in general it was well tolerated, some of the young participants commented that it was heavy. This indicates that further optimisation of weight is needed if we want to move towards younger (<2 years) participants. (Note that this is possible since, whilst the total weight is ~900 g, the combined sensor weight is just 250 g.) Similarly, here the warmth generated by the sensors was controlled by convection and insulation. However, for systems with a higher channel count, where more heat may be generated, active cooling (e.g. air forced through the helmet) may be required. Further, here magnetic field control (key to ensuring participants were unconstrained) was only available over a region encompassing the head whilst participants were seated (i.e. participants had to be sat in a chair for the scanner to work). However, in future studies, it may be desirable to accommodate different positions (e.g. participants seated on the floor or lying down) and a greater range of motion (e.g. crawling or walking). This may be possible with newly developing coil technology (*Holmes et al., 2023*).

## Neuroscientific insights

In addition to demonstrating a new platform for neurodevelopmental investigation, our study also provides insights into coordinated brain activity and its maturation with age. Beta oscillations are thought to mediate top-down influence on primary cortices, with regions of high beta amplitude being inhibited (for a review, see *Barone and Rossiter, 2021*). Whilst most evidence is based on studies of movement, there is significant supporting evidence from somatosensory studies in adults; e.g., *Bauer et al., 2014*, showed that, when one attends to events relating to the left hand, a relative decrease in beta amplitude is seen in the contralateral (right) sensory cortex and an increase in ipsilateral cortex – suggesting the brain is inhibiting the sensory representation of the non-relevant hand. Given this strong link to attentional mechanisms and top-down processing, it is unsurprising that beta oscillations are not fully developed in children, and consequently change with age.

The burst model of beta dynamics is relatively new, yet significant evidence already shows that the neurophysiological signal is driven by punctate bursts of pan-spectral activity, whose probability of occurrence changes depending on the task phase. Our study provides some of the first evidence (see also *Rayson et al., 2023*) that neurodevelopmental changes in the amplitude of task-induced beta modulation can also be explained by the burst model. Specifically, we showed that task-induced modulation of burst probability changes significantly as a function of age, suggesting bursts in somatosensory cortex are less likely to occur during stimulation of older participants compared to younger participants. We also showed that the 'classical' beta band modulation exhibited a significant linear relationship with burst probability modulation. In addition, when bursts occur in younger participants, they tend to have different spectral properties. Specifically, younger participants have increased low-frequency activity and decreased high-frequency activity, compared to adults. It is likely that a combination of the change in burst probability with age, and the change in dominant frequency (away from the canonical beta band), drives the observation from previous studies of changing beta modulation with age. Interestingly, we found no significant modulation of (broadband) burst amplitude with age. These findings are in good agreement with a recent paper which used EEG to probe burst modulation during observed movements in babies and adults (*Rayson et al., 2023*).

Our connectivity finding is also of note, showing a significant increase in functional connectivity with age. This is in good agreement with previous literature – e.g., *Schäfer et al., 2014*, showed quantitatively similar data in conventional MEG, albeit again by scanning older children (ages 6 and up). Here, we also showed that connectivity changes with age are most prominent in the frontal and parietal areas, and weakest in the visual cortex. It makes intuitive sense that the largest changes in connectivity over the age range studied should occur in the parietal and frontal regions – these regions are typically associated with both cognitive and attentional networks and are implicated in the networks that develop most between childhood and adulthood. Here, we observed a relative lack of age-related change in the visual regions; this could relate to the nature of the task – recall that all volunteers watched their favourite TV show and so the visual regions were being stimulated throughout, driving coordinated network activity in occipital cortex. The visual system is also early to mature compared to frontal cortex.

We failed to see a significant difference in the spatial location of the cortical representations of the index and little finger; there are three potential reasons for this. First, the system was not designed

to look for such a difference – sensors were sparsely distributed to achieve whole head coverage (rather than packed over sensory cortex to achieve the best spatial resolution in one area; *Hill et al., 2024*). Second, our 'pseudo-MRI' approach to head modelling (see Methods) is less accurate than acquisition of participant-specific MRIs, and so may mask subtle spatial differences. Third, we used a relatively straightforward technique for modelling magnetic fields generated by the brain (a single shell forward model). Although MEG is much less susceptible to conductivity inhomogeneity of the head than EEG, the forward model may still be impacted by the small head profile. This may diminish spatial resolution and future studies might look to implement more complex models based on, for example, finite element modelling (*Stenroos et al., 2014*). Finally, previous work (*Barratt et al., 2018*) suggested that, for a motor paradigm in adults, only the beta rebound, and not the power reduction during stimulation, mapped motortopically. This may also be the case for purely sensory stimulation. Nevertheless, it remains the case that by placing sensors closer to the scalp, OPM-MEG should offer improved spatial resolution in children and adults; this should be the topic of future work.

Finally, this was the first study of its kind using OPM-MEG, and consequently aspects of the study design could have been improved. Firstly, the task was designed for children; it was kept short while maximising the number of trials (to maximise signal-to-noise ratio). However, the classical view of beta modulation includes a PMBR which takes ~10 s to reach baseline following task cessation (*Pfurtscheller and Lopes da Silva, 1999*; *Fry et al., 2016*; *Pakenham et al., 2020*). Our short trial duration therefore doesn't allow the rebound to return to baseline between trials, and so conflates PMBR with rest. Consequently, we cannot differentiate the neural generators of the task-induced beta power decrease and the PMBR; whilst this helped ensure a short, child-friendly task, future studies should aim to use longer rest windows to independently assess which of the two processes is driving age-related changes. Secondly, here we chose to use passive (sensory) stimulation. This helped ensure compliance with the task in subjects of all ages and prevented confounds of, for example, reaction time, force, speed, and duration of movement which would be more likely in a motor task (*Fry et al., 2016*; *Pakenham et al., 2020*). However, there are many other systems to choose and whether the findings here regarding beta bursts and the changes with age also extend to other brain networks remains an open question. Thirdly, we lost more trials in children than we did in adults (19±12% compared to 9±5%) and this ostensibly implies a greater signal-to-noise ratio in adults compared to children which could help drive the effects observed. To test this, we ran a second analysis in which data were removed to equalise the final trial counts in the two groups (see Appendix 3). These additional analyses resulted in no change to our conclusions. Finally, the number of sensors available varied across participants – this was mainly for pragmatic purposes (the system was experimental and not all OPMs were available for every recording). Whilst we always ensured good coverage of sensorimotor cortex, and tried to optimise whole-brain coverage as much as we could, the system is likely to have diminished sensitivity around the temporal cortex, and this may explain why there was relatively little change in connectivity with age in those regions. In future, the inclusion of more sensors, particularly around the cheekbone, would be a natural extension.

## Conclusion

Characterising how neural oscillations change with age is a key step towards understanding the developmental trajectory of coordinated brain function, and the deviation of that trajectory in disorders. However, limitations of conventional, non-invasive approaches to measuring electrophysiology have led to confounds when scanning children. Here, we have demonstrated a new platform for neurodevelopmental assessment. Using OPM-MEG, we have been able to provide evidence – supported by previous studies – that shows both task-induced beta modulation and whole-brain functional connectivity increase with age. In addition, we have shown that the classically observed beta power drop during stimulation can be explained by a lower burst probability, and that modulation of burst probability changes with age. We further showed that the frequency content of bursts changes with age. Our results offer new insights into the developmental trajectory of beta oscillations and provide clear evidence that OPM-MEG is an ideal platform to study electrophysiology in neurodevelopment.

## Methods

### Participants and experiment

The study received ethical approval from the University of Nottingham Research Ethics Committee (Reference number 276-1802) and informed written consent, and consent to publish, was obtained from each participant, or where appropriate, the parents of the participants. Consent and authorisation for publication of *Figure 1A* were also obtained.

The paradigm comprised tactile stimulation of the tips of the index and little fingers using two braille stimulators (METEC, Germany) (see *Figure 1C*). Each stimulator comprised eight independently controlled pins which could be raised or lowered to tap the participant's finger. A single trial comprised approximately 0.5 s of stimulation during which the finger was tapped three times using all eight pins. Pins were up for 82 ms during each 'tap' and down for 82 ms between 'taps'. This was followed by 3 s rest. The finger stimulated (index or little) was alternated between trials. There was a total of 42 trials for each finger, meaning the experiment lasted a total of 294 s. Throughout the experiment, participants watched a television program of their choice (presented via projection onto a screen in the MSR, using a View Sonic PX748-4K projector at 60 Hz refresh rate). All children were accompanied by a parent and one experimenter throughout their visit.

### Data collection and co-registration

The sensor array comprised 64 triaxial OPMs (QuSpin Inc, Colorado, USA, Zero Field Magnetometer, Third Generation) which enabled a maximum of 192 measurements of magnetic field around the scalp (192 channels). The OPMs could be mounted in one of four 3D-printed helmets of different sizes (Cerca Magnetics Ltd., Nottingham, UK); this affords (approximate) whole-head coverage and adaptation to the participant's head size. All participants wore a thin aerogel cap underneath the helmet to control heat from the sensors (which operate with elevated temperature). The system is housed in an MSR equipped with degaussing coils (*Altarev et al., 2014*) and active magnetic field control (*Holmes et al., 2018*) (Cerca Magnetics Ltd., Nottingham, UK). Prior to data collection, the MSR was demagnetised and the magnetic field compensation coils energised (using currents based on previously obtained field maps). This procedure, which results in a background field of ~0.6 nT (*Rhodes et al., 2023*), is important to enable free head motion during a scan (*Borna et al., 2022*). All OPMs were equipped with on-board coils which were used for sensor calibration. MEG data were collected at a sampling rate of 1200 Hz (16-bit precision) using a National Instruments (NI, Texas, USA) data acquisition system interfaced with LabVIEW (NI).

Following data collection, two 3D digitisations of the participant's head, with and without the OPM helmet, were generated using a 3D structured light metrology scanner (Einscan H, SHINING 3D, Hangzhou, China). Participants wore a swimming cap to flatten hair during the 'head-only' scan. The head-only digitisation was used to measure head size and shape, and an age-matched T1-weighted template MRI scan was selected from a database (*Richards, 2019*) and warped to fit the digitisation, using FLIRT in FSL (*Jenkinson et al., 2002*; *Jenkinson and Smith, 2001*). This procedure resulted in a 'pseudo-MRI' which provided an approximation of the subject's brain anatomy. Sensor locations and orientations relative to this anatomy were found by aligning it to the digitisation of the participant wearing the sensor helmet, and adding the known geometry of the sensor locations and orientations within the helmet (*Zetter et al., 2019*; *Hill et al., 2020*; *Rier et al., 2023*). This was done using MeshLab (*Cignoni, 2008*).

### MEG data preprocessing

We used a preprocessing pipeline described previously (*Rier et al., 2023*). Briefly, broken or excessively noisy channels were identified by manual visual inspection of channel power spectra; any channels that were either excessively noisy, or had zero amplitude, were removed. Automatic trial rejection was implemented with trials containing abnormally high variance (exceeding 3 standard deviations from the mean) removed. All experimental trials were also inspected visually by an experienced MEG scientist, to exclude trials with large spikes/drifts that were missed by the automatic approach. In the adult group, there was a significant overlap between automatically and manually detected bad trials (0.7±1.6 trials were only detected manually). In the children 10.0±9.4 trials were only detected manually. Notch filters at the powerline frequency (50 Hz) and 2 harmonics, and a 1–150 Hz band pass filter, were applied. Finally, eye blink and cardiac artefacts were removed using ICA (implemented in

FieldTrip; *Oostenveld et al., 2011*) and homogeneous field correction was applied to reduce interference (*Tierney et al., 2021*).

## Source reconstruction and beta modulation

For source estimation, we used an LCMV beamformer spatial filter (*Van Veen et al., 1997*). For all analyses, covariance matrices were generated using data acquired throughout the whole experiment (excluding bad channels and trials). Covariance matrices were regularised using the Tikhonov method with a regularisation parameter equal to 5% of the maximum eigenvalue of the unregularised matrix. The forward model was based on a single shell volumetric conductor (*Nolte, 2003*).

To construct the *pseudo-T-statistical images*, data were filtered to the beta band (13–30 Hz) and narrow band data covariance matrices generated. Voxels were placed on both an isotropic 4 mm grid covering the whole brain and a 1 mm grid covering the contralateral sensorimotor regions. For each voxel, we contrasted power in active (0.3–0.8 s) and control (2.5–3 s) time windows to generate images showing the spatial signature of beta band modulation. Separate images were derived for index and little finger trials.

To generate *time-frequency spectra*, we used broadband (1–150 Hz) data and covariance matrices. The beamformer was used to produce a time course of neural activity (termed a 'virtual electrode') at the voxel with maximum beta band modulation (identified from the 1 mm resolution pseudo-T-statistical images). The resulting beamformer-projected broadband data were frequency filtered into a set of overlapping bands, and a Hilbert transform used to derive the analytic signal for each band. The absolute value of this was computed to give the envelope of oscillatory amplitude (termed the Hilbert envelope). This was averaged across trials, concatenated in frequency, baseline corrected, and normalised yielding a TFS showing relative change in spectral power (from baseline) as a function of time and frequency. To generate the *evoked response*, the broadband (4–40 Hz) beamformer-projected data (for the same location in sensorimotor cortex) were simply averaged across trials. To quantify the magnitude of *beta modulation*, we filtered the virtual electrode to the beta band, calculated the Hilbert envelope, averaged across trials and computed time courses of amplitude change relative to baseline (2.5–3 s). The beta modulation index ($\beta_{mod}$) was calculated using the equation $\beta_{mod} = (\beta_{Post} - \beta_{Stim})/\beta_{Baseline}$, where $\beta_{Stim}$, $\beta_{Post}$, and $\beta_{Baseline}$ are the average Hilbert-envelope-derived amplitudes in the stimulus (0.3–0.8 s), post-stimulus (1–1.5 s), and baseline (2.5–3 s) windows, respectively. To calculate the *evoked response amplitude*, we measured the amplitude of the evoked response at 50 ms post stimulation (the M50). These values (derived for every participant) were plotted against age and a relationship assessed via Pearson correlation.

## Functional connectivity analysis

To measure functional connectivity, we first parcellated the brain into distinct regions. To this end, estimated brain anatomies were co-registered to the MNI standard brain using FSL FLIRT (*Jenkinson et al., 2002*; *Jenkinson and Smith, 2001*) and divided into 78 cortical regions according to the automated anatomical labelling (AAL) atlas (*Tzourio-Mazoyer et al., 2002*; *Hillebrand et al., 2016*; *Gong et al., 2009*). Virtual electrode time courses were generated at the centre of mass of each of these 78 regions, and the beta band Hilbert envelope derived. We then computed AEC as an estimate of functional connectivity between all possible pairs of AAL regions (*Brookes et al., 2011a*; *O'Neill et al., 2015*). Prior to AEC, we applied pairwise orthogonalisation to reduce source leakage (*Brookes et al., 2012*; *Hipp et al., 2012*). This resulted in a single connectome matrix per participant. We estimated 'global connectivity' as the mean connectivity value across all off-diagonal elements in the connectome matrix. This was plotted against age and the relationship assessed using Pearson correlation. To visualise the spatial variation in age-related connectivity changes, we also estimated the correlation between node degree (i.e. the column-wise sum of connectome matrix elements) and age, for each of the 78 AAL regions.

## Beta bursts and HMM

To estimate beta burst timings we employed a three-state, time-delay embedded univariate HMM (*Vidaurre et al., 2016*). This method has been described extensively in previously papers (*Seedat et al., 2020*; *Rier et al., 2021*). Briefly, virtual electrode time series were frequency filtered 1–48 Hz. The HMM was used to divide this time course into three 'states' each depicting repeating patterns of

activity with similar temporo-spectral signatures. The output was three time courses representing the likelihood of each state being active as a function of time. These were binarised (using a threshold of 2/3) and used to generate measures of the probability of state occurrence as a function of time in a single trial. The state whose probability of occurrence modulated most with the task was defined as the 'burst state'. We estimated age-related changes in burst probability modulation and the relationship between burst probability modulation and classical beta modulation (see above) using Pearson correlation. Further, we investigated the spectral content of the burst state and its modulation with age using multi-taper estimation of the power spectral density (PSD) (*Vidaurre et al., 2016*). Having derived the spectral content of the burst state we used Pearson correlation to measure how the PSD magnitude, for every frequency, changes with age.

## Acknowledgements

This work was supported by an Engineering and Physical Sciences Research Council (EPSRC) Healthcare Impact Partnership Grant (EP/V047264/1) and an Innovate UK germinator award (Grant number 1003346). We also acknowledge support from the UK Quantum Technology Hub in Sensing and Timing, funded by EPSRC (EP/T001046/1). Sensor development was made possible by funding from the National Institutes of Health (R44MH110288).

## Additional information

### Competing interests

Lukas Rier: is a scientific advisor for Cerca Magnetics Limited, a company that sells equipment related to brain scanning using OPM-MEG. Elena Boto: is a director of Cerca Magnetics Limited, a company that sells equipment related to brain scanning using OPM-MEG. She also holds founding equity in Cerca Magnetics Limited. Niall Holmes, Ryan M Hill, Richard W Bowtell: is a scientific advisor for Cerca Magnetics Limited, a company that sells equipment related to brain scanning using OPM-MEG. He also holds founding equity in Cerca Magnetics Limited. Vishal Shah: is the founding director of QuSpin, a company that builds and sells OPM sensors. Cody Doyle, James Osborne: works for QuSpin, a company that builds and sells OPM sensors. Matthew J Brookes: is a director of Cerca Magnetics Limited, a company that sells equipment related to brain scanning using OPM-MEG. He also holds founding equity in Cerca Magnetics Limited. The other authors declare that no competing interests exist.

### Funding

| Funder | Grant reference number | Author |
| --- | --- | --- |
| Engineering and Physical Sciences Research Council | EP/V047264/1 | Richard W Bowtell Matthew J Brookes |
| Innovate UK | 1003346 | Elena Boto |
| Engineering and Physical Sciences Research Council | EP/T001046/1 | Richard W Bowtell Matthew J Brookes |
| National Institutes of Health | R44MH110288 | Vishal Shah |

The funders had no role in study design, data collection and interpretation, or the decision to submit the work for publication.

### Author contributions

Lukas Rier, Conceptualization, Data curation, Software, Formal analysis, Investigation, Visualization, Methodology, Writing – original draft, Writing – review and editing; Natalie Rhodes, Conceptualization, Data curation, Software, Formal analysis, Investigation, Visualization, Methodology, Project administration, Writing – review and editing; Daisie O Pakenham, Data curation, Investigation, Project administration, Writing – review and editing; Elena Boto, Conceptualization, Resources, Supervision, Methodology, Writing – review and editing; Niall Holmes, Software, Methodology, Writing – review

and editing; Ryan M Hill, Conceptualization, Software, Methodology, Writing – review and editing; Gonzalo Reina Rivero, Software, Writing – review and editing; Vishal Shah, Cody Doyle, James Osborne, Resources, Software, Writing – review and editing; Richard W Bowtell, Resources, Funding acquisition, Methodology, Writing – review and editing; Margot Taylor, Conceptualization, Supervision, Methodology, Writing – review and editing; Matthew J Brookes, Conceptualization, Resources, Supervision, Funding acquisition, Investigation, Methodology, Writing – original draft, Writing – review and editing

## Author ORCIDs

Lukas Rier https://orcid.org/0000-0002-4560-8787
Daisie O Pakenham https://orcid.org/0000-0002-0311-3102
Niall Holmes https://orcid.org/0000-0002-0561-8366
Matthew J Brookes https://orcid.org/0000-0002-8687-8185

## Ethics

Human subjects: The study received ethical approval from the University of Nottingham Research Ethics Committee (Reference number 276-1802) and informed written consent, and consent to publish, was obtained from each participant, or where appropriate, the parents of the participants. Consent and authorisation for publication of Figure 1A were also obtained.

Reviewer #3 (Public Review): https://doi.org/10.7554/eLife.94561.3.sa1
Author response https://doi.org/10.7554/eLife.94561.3.sa2

---

## Additional files

### Supplementary files

• MDAR checklist

### Data availability

All data used to produce the results presented here are made available on Zenodo. All code was made available on GitHub (copy archived at *Rier, 2024*).

The following dataset was generated:

| Author(s) | Year | Dataset title | Dataset URL | Database and Identifier |
|---|---|---|---|---|
| Rier L, Rhodes N, Pakenham D, Boto E, Holmes N, Hill RM, Reina Rivero G, Shah V, Doyle C, Osborne J, Bowtell R, Taylor M, Brookes M | 2024 | Tracking the neurodevelopmental trajectory of beta band oscillations with OPM-MEG (v1.0.0) [Data set] | https://doi.org/10.5281/zenodo.11126593 | Zenodo, 10.5281/zenodo.11126593 |

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

## Appendix 1

### Burst amplitude does not correlate with age

We showed a significant correlation between beta modulation and burst probability (*Figure 5D*) – implying that the stimulus-related drop in beta amplitude occurs because bursts are less likely to occur during this window. Further, we showed significant age-related changes in both beta amplitude modulation and burst probability, leading to a conclusion that the age dependence of beta modulation was caused by changes in the likelihood of bursts (i.e. bursts are less likely to 'switch off' during sensory stimulation, in children). Here, we extend these analyses to test whether burst amplitude also changes significantly with age. We reasoned that if burst amplitude remained the same in children and adults, this would not only suggest that beta modulation is driven solely by burst probability (distinct from children having lower amplitude bursts), but also show directly that the beta effects we see are not attributable to a lack of sensitivity in younger people.

We took the (unnormalised) beamformer-projected electrophysiological time series from sensorimotor cortex and filtered them 5–48 Hz. (The motivation for the large band was because bursts are known to be pan-spectral and have lower frequency content in children – this band captures most of the range of burst frequencies highlighted in our spectra.) We then extracted the timings of the bursts, and for each burst took the maximum projected signal amplitude. These values were averaged across all bursts in an individual subject and plotted for all subjects against age.

Results (see *Appendix 1—figure 1*) showed that the amplitude of the beta bursts showed no significant age-related modulation ($R^2=0.01$, p=0.48 for the index finger [*Appendix 1—figure 1A*] and $R^2=0.01$, p=0.57 for the little finger [*Appendix 1—figure 1B*]). This is distinct from both burst probability and task-induced beta modulation. This adds weight to the argument that the diminished beta modulation in children is not caused by a lack of sensitivity to the MEG signal and supports the conclusion that burst probability is the primary driver of age-related changes in beta oscillations.

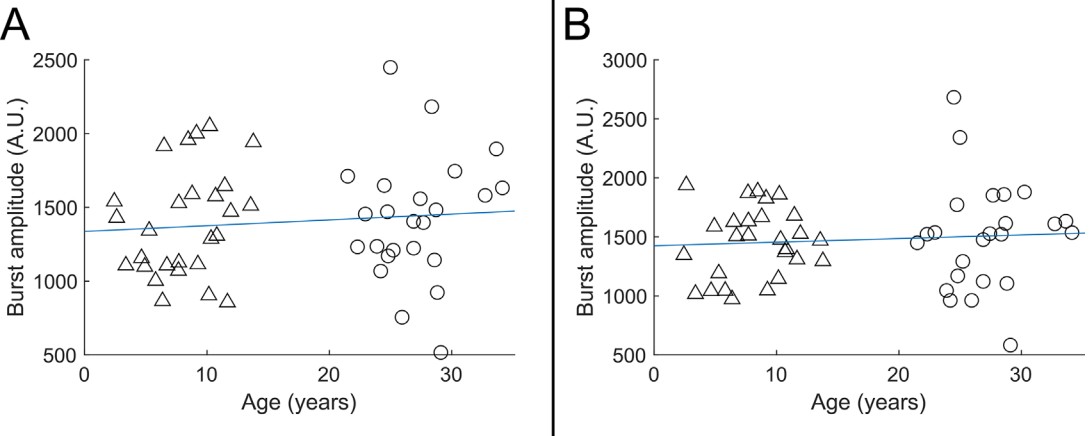

**Appendix 1—figure 1.** Beta burst amplitude as a function of age. **A** shows index finger simulation trials ($R^2=0.01$, p=0.48); **B** shows little finger stimulation trials ($R^2=0.01$, p=0.57). In both cases there was no significant modulation of burst amplitude with age.

# Appendix 2

## Proximity of sensors to the head

For an ideal wearable MEG system, the distance between the sensors and the scalp surface (sensor proximity) would be the same regardless of age (and head shape/size), ensuring maximum sensitivity in all subjects. To test how our system performed in this regard, we undertook analyses to compute scalp-to-sensor distances. This was done in two ways.

### Real distances in our adaptable system

We took the co-registered OPM sensor locations and computed the Euclidean distance from the centre of the sensitive volume (i.e. the centre of the vapour cell) to the closest point on the scalp surface. This was measured independently for all sensors, and an average across sensors was calculated. We repeated this for all participants (recall participants wore helmets of varying size and this adaptability should help minimise any relationship between sensor proximity and age).

### Simulated distances for a non-adaptable system

Here, the aim was to see how proximity might have changed with age, had only a single helmet size been used. We first identified the single example subject with the largest head (scanned wearing the largest helmet) and extracted the scalp-to-sensor distances as above. For all other subjects, we used a rigid body transform to co-register their brain to that of the example subject (placing their head [virtually] inside the largest helmet). Proximity was then calculated as above and an average across sensors calculated. This was repeated for all participants.

In both analyses, sensor proximity was plotted against age and significant relationships probed using Pearson correlation.

In addition, we also wanted to probe the relationship between sensor proximity and head circumference. Head circumference was estimated as follows: the whole-head MRI was binarised (to delineate the surface of the head); the axial slice with the largest area was selected and circumference of the head within that slice measured. We then plotted sensor proximity versus head circumference, for both the real (adaptive) and simulated (non-adaptive) case (expecting a negative relationship – i.e. larger heads mean closer sensor proximity). The slope of the relationship was measured and we used a permutation test to determine whether the use of adaptable helmets significantly lowered the identified slope (i.e. do adaptable helmets significantly improve sensor proximity in those with smaller head circumference).

Results are shown in *Appendix 2—figure 1*. We found no measurable relationship between sensor proximity and age ($r=-0.19$; $p=0.17$) in the case of the real helmets (panel A). When simulating a non-adaptable helmet, we did see a significant effect of age on scalp-to-sensor distance ($r=-0.46$; $p=0.001$; panel B). This demonstrates the advantage of the adaptability of OPM-MEG; without the ability to flexibly locate sensors, we would have a significant confound of sensor proximity.

Plotting sensor proximity against head circumference we found a significant negative relationship in both cases ($R=-0.37$; $p=0.007$ and $R=-0.78$; $p=0.000001$); however, the difference between slopes was significant according to a two-tailed permutation test ($p<0.025$), suggesting that adaptable helmets do indeed improve sensor proximity, in those with smaller head circumference. This again shows the benefits of adaptability to head size.

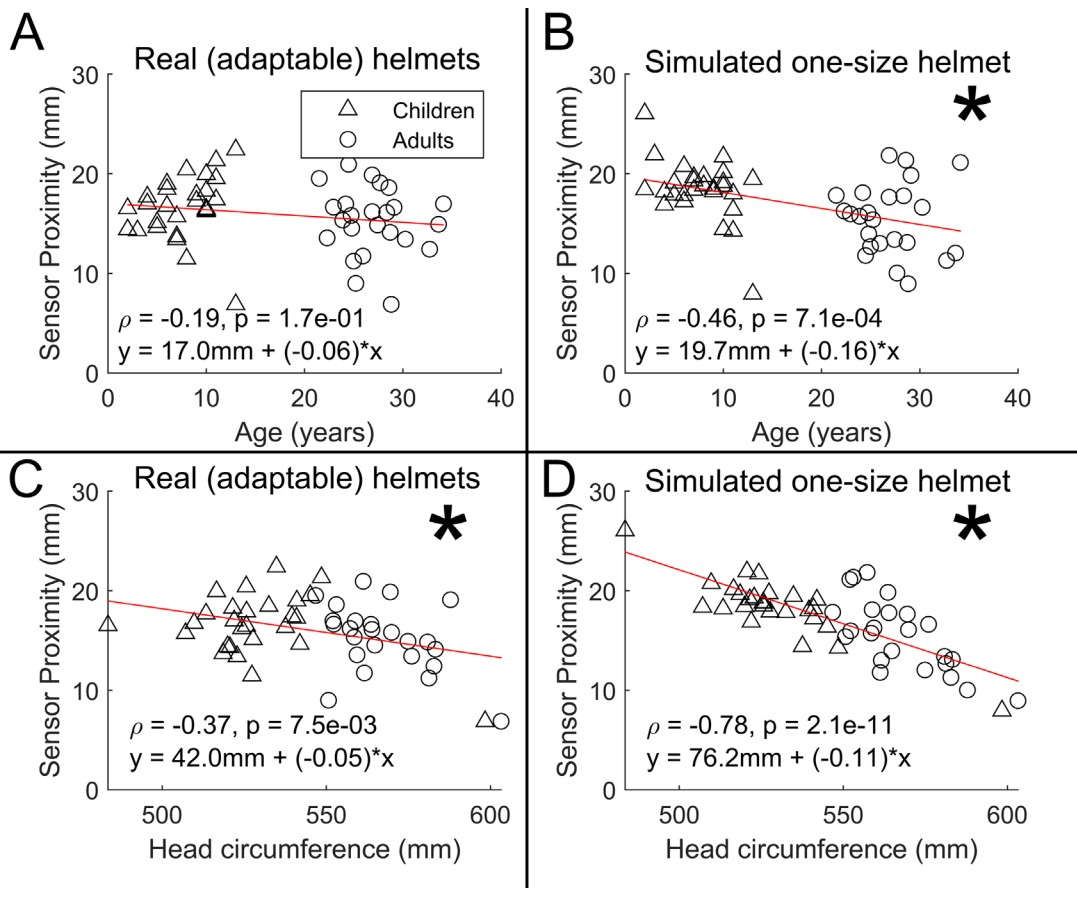

**Appendix 2—figure 1.** Scalp-to-sensor distance as a function of age (A/B) and head circumference (C/D). **A and C** show the case for the real helmets; **B and D** show the simulated non-adaptable case.

In sum, the ideal wearable system would see sensors located on the scalp surface, to get as close as possible to the brain in all subjects. Our system of multiple helmet sizes is not perfect in this regard (there is still a significant relationship between proximity and head circumference). However, our solution has offered a significant improvement over a (simulated) non-adaptable system. Future systems should aim to improve even further on this, either by using additively manufactured bespoke helmets for every subject (this is a gold standard, but also potentially costly for large studies), or adaptable flexible helmets.

## Appendix 3

### Reduced trial analyses

In our study, we had to discard more trials in children than adults. This potentially means a confound with a larger signal-to-noise ratio in adults than in children, which could affect the results. For this reason, we reanalysed our data, discarding trials from the adults to ensure equal numbers (on average) in our adult and child cohorts. Results are shown in *Appendix 3—figure 1*. Panel A shows beta modulation with age (equivalent to *Figure 3B*); panel B shows evoked response (M50) modulation with age (equivalent to *Figure 3C*); panel C shows functional connectivity with age (equivalent to *Figure 4B*); and panel D shows burst probability modulation with age (equivalent to *Figure 5C*). In all cases, the significant modulations with age captured in the main manuscript remain.

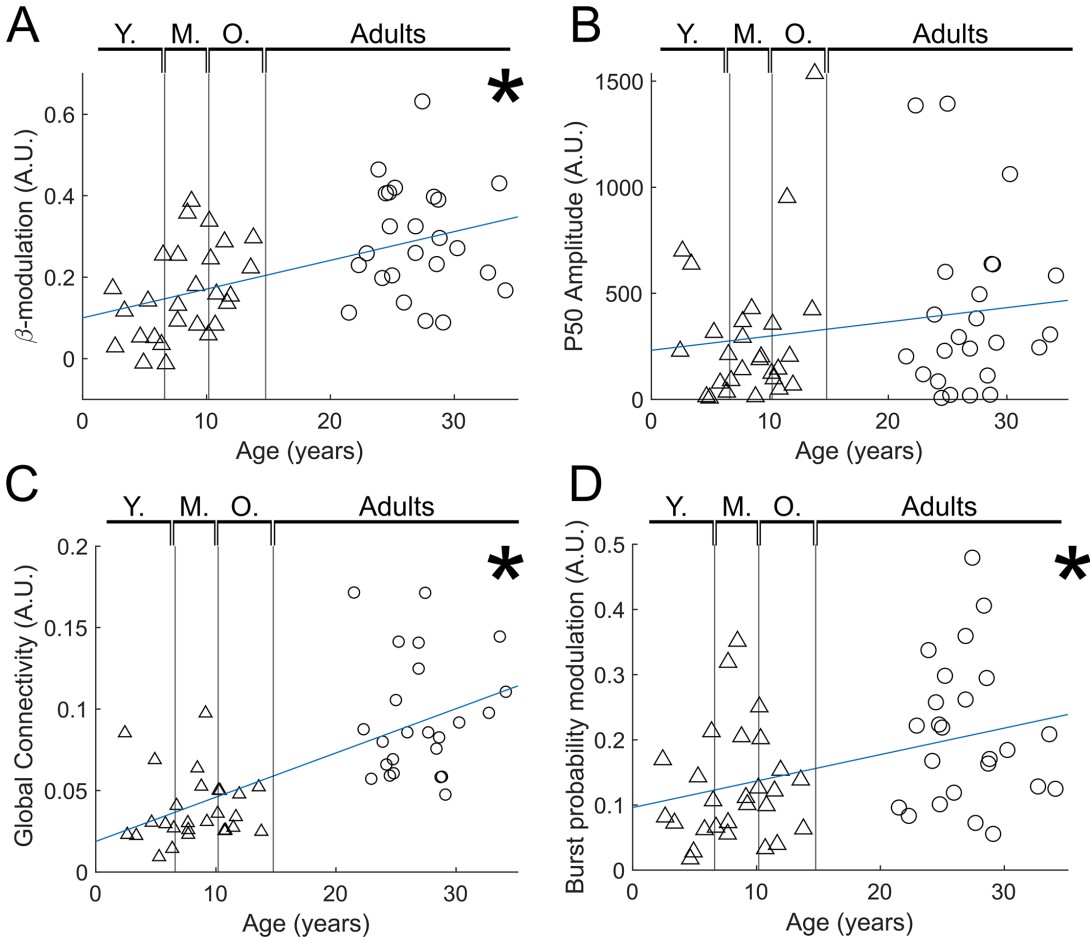

**Appendix 3—figure 1.** Reduced trial analysis for index finger stimuli. (**A**) Beta modulation with age ($R^2 = 0.26, p = 0.00014$). (**B**) Evoked response (P50) modulation with age ($R^2 = 0.03, p = 0.199$). (**C**) Functional connectivity with age ($R^2 = 0.45, p = 7 \times 10^{-8}$). (**D**) Burst probability modulation with age ( $R^2 = 0.15, p = 5.4 \times 10^{-3}$).

