## [Editor Report · eLife assessment]

This study provides **important** evidence supporting the ability of a new type of neuroimaging, OPM-MEG system, to measure beta-band oscillation in sensorimotor tasks in 2-14 years old children and to demonstrate the corresponding development changes, since neuroimaging methods with high spatiotemporal resolution that could be used on small children are quite limited. The evidence supporting the conclusion is **compelling**. This work will be of interest to the neuroimaging and developmental science communities.

---

## [Referee Report · Reviewer #3 (Public Review)]

This study demonstrated the application of OPM-MEG in neurodevelopment studies of somatosensory beta oscillations and connections with children as young as 2 years old. It provides a new functional neuroimaging method which has high spatial-temporal resolution as well wearable which makes it a new useful tool for studies in young children. They have constructed a 192-channel wearable OPM-MEG system includes field compensation coils which allows free head movement scanning with relatively high ratio of usable trials. Beta band oscillations during somatosensory tasks are well localized and the modulation with age are found in the amplitude, connectivity, and pan-spectral burst probability. It is demonstrated that the wearable OPM-MEG could be used in children as a quite practical and easy to deploy neuroimaging method with performance as good as conventional MEG. With both good spatial (several millimeter) and temporal (milliseconds) resolution, it provides a novel and powerful technology to neurodevelopment research and clinical application not limited to somatosensory areas.

The conclusions of this paper are mostly well supported by data acquired under proper method.

---

## [Author Response]

The following is the authors’ response to the original reviews.

**eLife assessment**

This study provides important evidence supporting the ability of a new type of neuroimaging, OPM-MEG system, to measure beta-band oscillation in sensorimotor tasks on 2-14 years old children and to demonstrate the corresponding development changes, since neuroimaging methods with high spatiotemporal resolution that could be used on small children are quite limited. The evidence supporting the conclusion is solid but lacks clarifications about the much-discussed advantages of OPM-MEG system (e.g., motion tolerance), control analyses (e.g., trial number), and rationale for using sensorimotor tasks. This work will be of interest to the neuroimaging and developmental science communities.

We thank the editors and reviewers for their time and comments on our manuscript. We have responded in detail to the comments, on a point-by-point basis, below. Included in our responses (and our revised manuscript) are additional analyses to control for trial count, clarification of the advantages of OPM-MEG, and justification of our use of sensory (as distinct from motor) stimulation. In what follows, our responses are in bold typeface; additions to our manuscript are in bold italic typeface.

**Reviewer #1 (Public Review):**
Summary:Compared with conventional SQUID-MEG, OPM-MEG offers theoretical advantages of sensor configurability (that is, sizing to suit the head size) and motion tolerance (the sensors are intrinsically in the head reference frame). This study purports to be the first to experimentally demonstrate these advantages in a developmental study from age 2 to age 34. In short, while the theoretical advantages of OPM-MEG are attractive - both in terms of young child sensitivity and in terms of motion tolerance - neither was in fact demonstrated in this manuscript. We are left with a replication of SQUID-MEG observations, which certainly establishes OPM-MEG as "substantially equivalent" to conventional technology but misses the opportunity to empirically demonstrate the much-discussed theoretical advantages/opportunities.

Thank you for reviewing our manuscript. We agree that our results demonstrate substantial equivalence with conventional MEG. However, as mentioned by Reviewer 3, most past studies have “focused on older children and adolescents (e.g., 9-15 years old)” whereas our youngest group is 25 years. We believe that by obtaining data of sufficient quality in these age groups, without the need for any restriction of head movement, we have demonstrated the advantage of OPM-MEG. We now have made this clear in our discussion:

“…our primary aim was to test the feasibility of OPM-MEG for neurodevelopmental studies. Our results demonstrate we were able to scan children down to age 2 years, measuring high-fidelity electrophysiological signals and characterising the neurodevelopmental trajectory of beta oscillations. The fact that we were able to complete this study demonstrates the advantages of OPM-MEG over conventional-MEG, the latter being challenging to deploy across such a large age range…”

Strengths:A replication of SQUID-MEG observations, which certainly establishes OPM-MEG as "substantially equivalent" to conventional technology but misses the opportunity to empirically demonstrate the much-discussed theoretical advantages/opportunities.

As noted above the demonstration of equivalence was one of our primary aims. We have elaborated further on the advantages below.

Weaknesses:The authors describe 64 tri-axial detectors, which they refer to as 192 channels. This is in keeping with some of the SQUID-MEG description, but possibly somewhat disingenuous. For the scientific literature, perhaps "64 tri-axial detectors" is a more parsimonious description.

The number of channels in a MEG system refers to the number of independent measurements of magnetic field. This, in turn, tells us the number of degrees of freedom in the data that can be exploited by algorithms like signal space separation or beamforming. E.g. the MEGIN (cryogenic) MEG system has 306 channels, 102 magnetometers and 204 planar gradiometers. Sensors are constructed as “triple sensor elements” with one magnetometer and 2 gradiometers (in orthogonal orientations) centred on a single location. In our system, each sensor has three orthogonal metrics of magnetic field which are (by definition) independent. We have 64 such sensors, and therefore 192 independent channels – indeed when implementing algorithms like SSS we have shown we can exploit this number of degrees of freedom (1). 192 channels is therefore an accurate description of the system.

A small fraction (<20%) of trials were eliminated for analysis because of "excess interference" - this warrants further elaboration.

We agree that this is an important point. We now state in our methods section:

“…Automatic trial rejection was implemented with trials containing abnormally high variance (exceeding 3 standard deviations from the mean) removed. All experimental trials were also inspected visually by an experienced MEG scientist, to exclude trials with large spikes/drifts that were missed by the automatic approach. In the adult group, there was a significant overlap between automatically and manually detected bad trials (0.7+-1.6 trials were only detected manually). In the children (10.0 +-9.4 trials were only detected manually)…”

We also note that the other reviewers and editor questioned whether the higher rejection rate in children had any bearing on results. This is an extremely important question. In revising the manuscript this has also been taken into account with all data reanalysed with equal trial counts in children and adults. Results are presented in Supplementary Information Section 5.

Figure 3 shows a reduced beta ERD in the youngest children. Although the authors claim that OPMMEG would be similarly sensitive for all ages and that SQUID-MEG would be relatively insensitive to young children, one trivial counterargument that needs to be addressed is that OPM has NOT in fact increased the sensitivity to young child ERD. This can possibly be addressed by analogous experiments using a SQUID-based system. An alternative would be to demonstrate similar sensitivity across ages using OPM to a brain measure such as evoked response amplitude. In short, how does Figure 3 demonstrate the (theoretical) sensitivity advantage of OPM MEG in small heads ?

We completely understand the referees’ point – indeed the question of whether a neuromagnetic effect really changes with age, or apparently changes due to a drop in sensitivity (caused by reduced head size or - in conventional MEG and fMRI - increased subject movement) is a question that can be raised in all neurodevelopmental studies.

Our authors have many years’ experience conducting studies using conventional MEG (including in neurodevelopment) and agreed that the idea of scanning subjects down to age two in conventional MEG would not be practical; their heads are too small and they typically fail to tolerate an environment where they are forced to remain still for long periods. Even if we tried a comparative study using conventional MEG, the likely data exclusion rate would be so high that the study would be confounded. This is why most conventional MEG studies only scan older children and adolescents. For this reason, we cannot undertake the comparative study the reviewer suggests. There are however two reasons why we believe sensitivity is not driving the neurodevelopmental effects that we observe:

Proximity of sensors to the head:

For an ideal wearable MEG system, the distance between the sensors and the scalp surface (sensor proximity) would be the same regardless of age (and size), ensuring maximum sensitivity in all subjects. To test how our system performed in this regard, we undertook analyses to compute scalp-to-sensor distances. This was done in two ways:

(1) Real distances in our adaptable system: We took the co-registered OPM sensor locations and computed the Euclidean distance from the centre of the sensitive volume (i.e. the centre of the vapour cell) to the closest point on the scalp surface. This was measured independently for all sensors, and an average across sensors calculated. We repeated this for all participants (recall participants wore helmets of varying size and this adaptability should help minimise any relationship between sensor proximity and age).

(2) Simulated distances for a non-adaptable system: Here, the aim was to see how proximity might have changed with age, had only a single helmet size been used. We first identified the single example subject with the largest head (scanned wearing the largest helmet) and extracted the scalpto-sensor distances as above. For all other subjects, we used a rigid body transform to co-register their brain to that of the example subject (placing their head (virtually) inside the largest helmet). Proximity was then calculated as above and an average across sensors calculated. This was repeated for all participants.

In both analyses, sensor proximity was plotted against age and significant relationships probed using Pearson correlation.

In addition, we also wanted to probe the relation between sensor proximity and head circumference. Head circumference was estimated by binarising the whole-head MRI (to delineate volume of the head), and the axial slice with the largest circumference around was selected. We then plotted sensor proximity versus head circumference, for both the real (adaptive) and simulated (nonadaptive) case (expecting a negative relationship – i.e. larger heads mean closer sensor proximity). The slope of the relationship was measured and we used a permutation test to determine whether the use of adaptable helmets significantly lowered the identified slope (i.e. do adaptable helmets significantly improve sensor proximity in those with smaller head circumference).

Results are shown in Figure R1. We found no measurable relationship between sensor proximity and age (r = -0.195; p = 0.171) in the case of the real helmets (panel A). When simulating a non-adaptable helmet, we did see a significant effect of age on scalp-to-sensor distance (r = -0.46; p = 0.001; panel B). This demonstrates the advantage of the adaptability of OPM-MEG; without the ability to flexibly locate sensors, we would have a significant confound of sensor proximity.

Plotting sensor proximity against head circumference we found a significant negative relationship in both cases (r = -0.37; p = 0.007 and r = -0.78; p = 0.000001); however, the difference between slopes was significant according to a permutation test (p < 0.025) suggesting that adaptable has indeed improved sensor proximity in those with smaller head circumference. This again shows the benefits of adaptability to head size.

**Author response image 1. sa2fig1:** Scalp-to-sensor distance as a function of age (**A/B**) and head circumference (**C/D**). **A** and **C** show the case for the real helmets; **B** and **D** show the simulated non-adaptable case.

In sum, the ideal wearable system would see sensors located on the scalp surface, to get as close as possible to the brain in all subjects. Our system of multiple helmet sizes is not perfect in this regard (there is still a significant relationship between proximity and head circumference). However, our solution has offered a significant improvement over a (simulated) non-adaptable system. Future systems should aim to improve even further on this, either by using additively manufactured bespoke helmets for every subject (this is a gold standard, but also costly for large studies), or potentially adaptable flexible helmets.

Burst amplitudes:

The reviewer suggested to “demonstrate similar sensitivity across ages using OPM to a brain measure”. We decided not to use the evoked response amplitude (as suggested), since this would be expected to change with age. Instead, we used the amplitude of the bursts.

Our manuscript shows a significant correlation between beta modulation and burst probability – implying that the stimulus-related drop in beta amplitude occurs because bursts are less likely to occur. Further, we showed significant age-related changes in both beta amplitude and burst probability leading to a conclusion that the age dependence of beta modulation was caused by changes in the likelihood of bursts (i.e. bursts are less likely to ’switch off’ during sensory stimulation in children). We have now extended these analyses to test whether burst amplitude also changes significantly with age – we reasoned that if burst amplitude remained the same in children and adults, this would not only suggest that beta modulation is driven by burst probability (distinct from burst amplitude), but also show directly that the beta effects we see are not attributable to a lack of sensitivity in younger people.

We took the (unnormalized) beamformer projected electrophysiological time series from sensorimotor cortex and filtered it 5-48 Hz (the motivation for the large band was because bursts are known to be pan-spectral and have lower frequency content in children; this band captures most of the range of burst frequencies highlighted in our spectra). We then extracted the timings of the bursts, and for each burst took the maximum projected signal amplitude. These values were averaged across all bursts in an individual subject, and plotted for all subjects against age.

**Author response image 2. sa2fig2:** Beta burst amplitude as a function of age; (**A**) shows index finger simulation trials; (**B**) shows little finger stimulation trials. In both case there was no significant modulation of burst amplitude with age.

Results (see Figure R2) showed that the amplitude of the beta burst showed no significant age-related modulation (R^2^ = 0.01, p = 0.48 for index finger and R^2^ = 0.01, p = 0.57 for the little finger). This is distinct from both burst probability and task induced beta modulation. This adds weight to the argument that the diminished beta modulation in children is not caused by a lack of sensitivity to the MEG signal and supports our conclusion that burst probability is the primary driver of the agerelated changes in beta oscillations.

Both of the above analyses have been added to our supplementary information and mentioned in the main manuscript. The first shows no confound of sensor proximity to the scalp with age in our study. The second shows that the bursts underlying the beta signal are not significantly lower amplitude in children – which we reasoned they would be if sensitivity was diminished at younger ages. We believe that the two together suggest that we have mitigated a sensitivity confound in our study.

The data do not make a compelling case for the motion tolerance of OPM-MEG. Although an apparent advantage of a wearable system, an empirical demonstration is still lacking. How was motion tracked in these participants?

We agree that this was a limitation of our experiment.

We have the equipment to track motion of the head during an experiment, using IR retroreflective markers placed on the helmet and a set of IR cameras located inside the MSR. However, the process takes a long time to set up, it lacks robustness, and would have required an additional computer (the one we typically use was already running the somatosensory stimulus and video). When the study was designed, we were concerned that the increased set up time for motion tracking would cause children to get bored, and result in increased participant drop out. For this reason we decided not to capture motion of the head during this study.

With hindsight this was a limitation which – as the reviewer states – makes us unable to prove that motion robustness was a significant advantage for this study. That said, during scanning there was both a parent and an experimenter in the room for all of the children scanned, and anecdotally we can say that children tended to move their head during scans – usually to talk to the parent. Whilst this cannot be quantified (and is therefore unsatisfactory) we thought it worth mentioning in our discussion, which reads:

“…One limitation of the current study is that practical limitations prevented us from quantitatively tracking the extent to which children (and adults) moved their head during a scan. Anecdotally however, experimenters present in the room during scans reported several instances where children moved, for example to speak to their parents who were also in the room. Such levels of movement could not be tolerated in conventional MEG or MRI and so this again demonstrates the advantages afforded by OPM-MEG…”

As a note, empirical demonstrations of the motion tolerance of OPM-MEG have been published previously: Early demonstrations included Boto et al (2). who captured beta oscillations in adults playing a ball game and Holmes et al. who measured visual responses as participants moved their head to change viewing angle (3). In more recent demonstrations, Seymour et al. measured the auditory evoked field in standing mobile participants (4); Rea et al. measured beta modulation as subjects carried out a naturalistic handwriting task5 and Holmes et al measured beta modulation as a subject walked around a room (6).

Furthermore, while the introduction discusses at some length the phenomenon of PMBR, there is no demonstration of the recording of PMBR (or post-sensory beta rebound). This is a shame because there is literature suggesting an age-sensitivity to this, that the optimal sensitivity of OPM-MEG might confirm/refute. There is little evidence in Figure 3 for adult beta rebound. Is there an explanation for the lack of sensitivity to this phenomenon in children/adolescents? Could a more robust paradigm (button-press) have shed light on this?

We understand the question. There are two limitations to the current study in respect to measuring the PMBR:

Firstly, sensory tasks generally do not induce as strong a PMBR as motor tasks and with this in mind a stronger rebound response could have been elicited using a button press. However, it was our intention to scan children down to age 2 and we were sceptical that the youngest children would carry out a button press as instructed. For this reason we opted for entirely passive stimulation, requiring no active engagement from our participants. The advantages of this was a stimulus that all subjects could engage with. However, this was at the cost of a diminished rebound.

The second limitation relates to trial length. Multiple studies have shown that the PMBR can last over ~10 s (7,8). Indeed, Pfurtscheller et al. argued in 1999 that it was necessary to leave 10 s between movements to allow the PMBR to return to a true baseline (9), though this has rarely been adhered to in the literature. Here, we wanted to keep recordings short for the comfort of the younger participants, so we adopted a short trial duration. However, a consequence of this short trial length is that it becomes impossible to access the PMBR directly; one can only measure beta modulation with the task. This limitation has now been addressed explicitly in our discussion:

“…this was the first study of its kind using OPM-MEG, and consequently aspects of the study design could have been improved. Firstly, the task was designed for children; it was kept short while maximising the number of trials (to maximise signal to noise ratio). However, the classical view of beta modulation includes a PMBR which takes ~10 s to reach baseline following task cessation (7–9). Our short trial duration therefore doesn’t allow the rebound to return to baseline between trials, and so conflates PMBR with rest. Consequently, we cannot differentiate the neural generators of the task induced beta power decrease and the PMBR; whilst this helped ensure a short, child friendly task, future studies should aim to use longer rest windows to independently assess which of the two processes is driving age related changes…”

Data on functional connectivity are valuable but do not rely on OPM recording. They further do not add strength to the argument that OPM MEG is more sensitive to brain activity in smaller heads - in fact, the OPM recordings seem plagued by the same insensitivity observed using conventional systems.

Given the demonstration above that bursts are not significantly diminished in amplitude in children relative to adults; and further given the demonstrations in the literature (e.g. Seedat et al (10).) that functional connectivity is driven by bursts, we would argue that the effects of connectivity changing with age are not related to sensitivity but rather genuinely reflect a lack of coordination of brain activity.

The discussion of burst vs oscillations, while highly relevant in the field, is somewhat independent of the OPM recording approach and does not add weight to the OPM claims.

We agree that the burst vs. oscillations discussion does not add weight to the OPM claims per se. However, we had two aims of our paper, the second being to “investigate how task-induced beta modulation in the sensorimotor cortices is related to the occurrence of pan-spectral bursts, and how the characteristics of those bursts change with age.” As the reviewer states, this is highly relevant to the field, and therefore we believe adds impact, not only to the paper, but also by extension to the technology.

In short, while the theoretical advantages of OPM-MEG are attractive - both in terms of young child sensitivity and in terms of motion tolerance, neither was in fact demonstrated in this manuscript. We are left with a replication of SQUID-MEG observations, which certainly establishes OPM-MEG as "substantially equivalent" to conventional technology but misses the opportunity to empirically demonstrate the much-discussed theoretical advantages/opportunities.

We thank the referee for the time and important contributions to this paper. We believe the fact that we were able to record good data in children as young as two years old was, in itself, an experimental realisation of the ‘theoretical advantages’ of OPM-MEG. Our additional analyses, inspired by the reviewers comments, help to clarify the advantages of OPM-MEG over conventional technology. The reviewers’ insights have without doubt improved the paper.

**Reviewer #2 (Public Review):**
Summary:The authors introduce a new 192-channel OPM system that can be configured using different helmets to fit individuals from 2 to 34 years old. To demonstrate the veracity of the system, they conduct a sensorimotor task aimed at mapping developmental changes in beta oscillations across this age range. Many past studies have mapped the trajectory of beta (and gamma) oscillations in the sensorimotor cortices, but these studies have focused on older children and adolescents (e.g., 9-15 years old) and used motor tasks. Thus, given the study goals, the choice of a somatosensory task was surprising and not justified. The authors recorded a final sample of 27 children (2-13 years old) and 24 adults (21-34 years) and performed a time-frequency analysis to identify oscillatory activity. This revealed strong beta oscillations (decreases from baseline) following the somatosensory stimulation, which the authors imaged to discern generators in the sensorimotor cortices. They then computed the power difference between 0.3-0.8 period and 1.0-1.5 s post-stimulation period and showed that the beta response became stronger with age (more negative relative to the stimulation period). Using these same time windows, they computed the beta burst probability and showed that this probability increased as a function of age. They also showed that the spectral composition of the bursts varied with age. Finally, they conducted a whole-brain connectivity analysis. The goals of the connectivity analysis were not as clear as prior studies of sensorimotor development have not conducted such analyses and typically such whole-brain connectivity analyses are performed on resting-state data, whereas here the authors performed the analysis on task-based data. In sum, the authors demonstrate that they can image beta oscillations in young children using OPM and discern developmental effects.

Thank you for this summary and for taking the time to review our manuscript.

Strengths:Major strengths of the study include the novel OPM system and the unique participant population going down to 2-year-olds. The analyses are also innovative in many respects.

Thank you – we also agree that the major strength is in the unique cohort.

Weaknesses:Several weaknesses currently limit the impact of the study.First, the choice of a somatosensory stimulation task over a motor task was not justified. The authors discuss the developmental motor literature throughout the introduction, but then present data from a somatosensory task, which is confusing. Of note, there is considerable literature on the development of somatosensory responses so the study could be framed with that.

We completely understand the referee’s point, and we agree that the motivation for the somatosensory task was not made clear in our original manuscript.

Our choice of task was motivated completely by our targeted cohort; whilst a motor task would have been our preference, it was generally felt that making two-year-olds comply with instructions to press a button would have been a significant challenge. In addition, there would likely have been differences in reaction times. By opting for a passive sensory stimulation we ensured compliance, and the same stimulus for all subjects. We have added text on this to our introduction as follows:

“…Here, we combine OPM-MEG with a burst analysis based on a Hidden Markov Model (HMM) (10–12) to investigate beta dynamics. We scanned a cohort of children and adults across a wide age range (upwards from 2 years old). Because of this, we implemented a passive somatosensory task which can be completed by anyone, regardless of age…”

We also state in our discussion:

“…here we chose to use passive (sensory) stimulation. This helped ensure compliance with the task in subjects of all ages and prevented confounds of e.g. reaction time, force, speed and duration of movement which would be more likely in a motor task (7,8). However, there are many other systems to choose and whether the findings here regarding beta bursts and the changes with age also extend to other brain networks remains an open question.…”

Regarding the neurodevelopmental literature – we are aware of the literature on somatosensory evoked responses – particularly median nerve stimulation – but we can find little on the neurodevelopmental trajectory of somatosensory induced beta oscillations (the topic of our paper). We have edited our introduction as follows:

“…All these studies probed beta responses to movement execution; in the case of tactile stimulation (i.e. sensory stimulation without movement) both task induced beta power loss, and the post stimulus rebound have been consistently observed in adults (9,13–18). Further, beta amplitude in sensory cortex has been related to attentional processes (19) and is broadly thought to carry top down top down influence on primary areas (20). However, there is less literature on how beta modulation changes with age during purely sensory tasks.…”

We would be keen for the reviewer to point to any specific papers in the literature that we may have missed.

Second, the primary somatosensory response actually occurs well before the time window of interest in all of the key analyses. There is an established literature showing mechanical stimulation activates the somatosensory cortex within the first 100 ms following stimulation, with the M50 being the most robust response. The authors focus on a beta decrease (desynchronization) from 0.3-0.8 s which is obviously much later, despite the primary somatosensory response being clear in some of their spectrograms (e.g., Figure 3 in older children and adults). This response appears to exhibit a robust developmental effect in these spectrograms so it is unclear why the authors did not examine it. This raises a second point; to my knowledge, the beta decrease following stimulation has not been widely studied and its function is unknown. The maps in Figure 3 suggest that the response is anterior to the somatosensory cortex and perhaps even anterior to the motor cortex. Since the goal of the study is to demonstrate the developmental trajectory of well-known neural responses using an OPM system, should the authors not focus on the best-understood responses (i.e., the primary somatosensory response that occurs from 0.0-0.3 s)?

We understand the reviewer’s point. The original aim of our manuscript was to investigate the neurodevelopmental trajectory of beta oscillations, not the evoked response. In fact, the evoked response in this paradigm is complicated by the fact that there are three stimuli in a very short (<500 ms) time window. For this reason, we prefer the focus of our paper to remain on oscillations.

Nevertheless, we agree that not including the evoked responses was a missed opportunity. We have now added evoked responses to our analysis pipeline and manuscript. As surmised by the reviewer, the M50 shows neurodevelopmental changes (an increase with age). Our methods section has been updated accordingly and Figure 3 has been modified. The figure and caption are copied below for the convenience of the reviewer.

**Author response image 3. sa2fig3:** Beta band modulation with age. (**A**) Brain plots show slices through the left motor cortex, with a pseudo-T-statistical map of beta modulation (blue/green) overlaid on the standard brain. Peak MNI coordinates are indicated for each subgroup. Time-frequency spectrograms show modulation of the amplitude of neural oscillations (fractional change in spectral amplitude relative to the baseline measured in the 2.5-3 s window). Vertical lines indicate the time of the first braille stimulus. In all cases results were extracted from the location of peak beta desynchronisation (in the left sensorimotor cortex). Note the clear beta amplitude reduction during stimulation. The inset line plots show the 4-40 Hz trial averaged phase-locked evoked response, with the expected prominent deflections around 20 and 50 ms. (**B**) Maximum difference in beta-band amplitude (0.3-0.8 s window vs 1-1.5 s window) plotted as a function of age (i.e., each data point shows a different participant; triangles represent children, circles represent adults). Note significant correlation (R^2^ = 0.29, p = 0.00004 *). (**C**) Amplitude of the P50 component of the evoked response plotted against age. There was no significant correlation (R^2^ = 0.04, p = 0.14). All data here relate to the index finger stimulation; similar results are available for the little finger stimulation in Supplementary Information Section 1.

Regarding the developmental effects, the authors appear to compute a modulation index that contrasts the peak beta window (.3 to .8) to a later 1.0-1.5 s window where a rebound is present in older adults. This is problematic for several reasons. First, it prevents the origin of the developmental effect from being discerned, as a difference in the beta decrease following stimulation is confounded with the beta rebound that occurs later. A developmental effect in either of these responses could be driving the effect. From Figure 3, it visually appears that the much later rebound response is driving the developmental effect and not the beta decrease that is the primary focus of the study. Second, these time windows are a concern because a different time window was used to derive the peak voxel used in these analyses. From the methods, it appears the image was derived using the .3-.8 window versus a baseline of 2.5-3.0 s. How do the authors know that the peak would be the same in this other time window (0.3-0.8 vs. 1.0-1.5)? Given the confound mentioned above, I would recommend that the authors contrast each of their windows (0.3-0.8 and 1.0-1.5) with the 2.5-3.0 window to compute independent modulation indices. This would enable them to identify which of the two windows (beta decrease from 0.3-0.8 s or the increase from 1.0-1.5 s) exhibited a developmental effect. Also, for clarity, the authors should write out the equation that they used to compute the modulation index. The direction of the difference (positive vs. negative) is not always clear.

We completely understand the referee’s point; referee 1 made a similar point. In fact, there are two limitations of our paradigm regarding the measurement of PMBR versus the task-induced beta decrease:

Firstly, sensory tasks generally do not induce as strong a PMBR as motor tasks and with this in mind a stronger rebound response could have been elicited using a button press. However, as described above it was our intention to scan children down to age 2 and we were sceptical that the youngest children would carry out a button press as instructed.

The second limitation relates to trial length. Multiple studies have shown that the PMBR can last over ~10 s (7,8). Indeed, Pfurtscheller et al. argued in 1999 that it was necessary to leave 10 s between movements to allow the PMBR to return to a true baseline (9) Here, we wanted to keep recordings relatively short for the younger participants, and so we adopted a short trial duration. However, a consequence of this short trial length is that it becomes impossible to access the PMBR directly because the PMBR of the nth trial is still ongoing when the (n+1)th trial begins. Because of this, there is no genuine rest period, and so the stimulus induced beta decrease and subsequent rebound cannot be disentangled. This limitation has now been made clear in our discussion as follows:

“…this was the first study of its kind using OPM-MEG, and consequently aspects of the study design could have been improved. Firstly, the task was designed for children; it was kept short while maximising the number of trials (to maximise signal to noise ratio). However, the classical view of beta modulation includes a PMBR which takes ~10 s to reach baseline following task cessation (7–9). Our short trial duration therefore doesn’t allow the rebound to return to baseline between trials, and so conflates PMBR with rest. Consequently, we cannot differentiate the neural generators of the task induced beta power decrease and the PMBR; whilst this helped ensure a short, child friendly task, future studies should aim to use longer rest windows to independently assess which of the two processes is driving age related changes…”

To clarify our method of calculating the modulation index, we have added the following statement to the methods:

“The beta modulation index (βmod ) was calculated using the equation βmod =(βPost −βttim )/βBaseline , where βstim ,βPost , and βBaseline  are the average Hilbert-envelope-derived amplitudes in the stimulus (0.3-0.8s), post-stimulus (1-1.5s) and baseline (2.5-3s) windows, respectively.”

Another complication of using a somatosensory task is that the literature on bursting is much more limited and it is unclear what the expectations would be. Overall, the burst probability appears to be relatively flat across the trial, except that there is a sharp decrease during the beta decrease (.3-.8 s). This matches the conventional trial-averaging analysis, which is good to see. However, how the bursting observed here relates to the motor literature and the PMBR versus beta ERD is unclear.

Again, we agree completely; a motor task would have better framed the study in the context of existing burst literature – but as mentioned above, making 2-year-olds comply with the instructions for a motor task would have been difficult. Interestingly in a recent paper, Rayson et al. used EEG to investigate burst activity in infants (9 and 12 months) and adults during observed movement execution, with results showing stimulus induced decrease in beta burst rate at all ages, with the largest effects in adults (21). This paper was not yet published when we submitted our article but does help us to frame our burst results since there is strong agreement between their study and ours. We now mention this study in both our introduction and discussion.

Another weakness is that all participants completed 42 trials, but 19% of the trials were excluded in children and 9% were excluded in adults. The number of trials is proportional to the signal-to-noise ratio. Thus, the developmental differences observed in response amplitude could reflect differences in the number of trials that went into the final analyses.

This is an important observation and we thank the reviewer for raising the issue. We have now re-analysed all of our data, removing trials in the adults such that the overall number of trials was the same as for the children. All effects with age remained significant. We chose to keep the Figures in the main manuscript with all good trials (as previously) and present the additional analyses (with matched trial numbers) in supplementary information. However, if the reviewer feels strongly, we could do it the other way around (there is very little difference between the results).

**Reviewer #3 (Public Review):**
This study demonstrated the application of OPM-MEG in neurodevelopment studies of somatosensory beta oscillations and connections with children as young as 2 years old. It provides a new functional neuroimaging method that has a high spatial-temporal resolution as well wearable which makes it a new useful tool for studies in young children. They have constructed a 192-channel wearable OPM-MEG system that includes field compensation coils which allow free head movement scanning with a relatively high ratio of usable trials. Beta band oscillations during somatosensory tasks are well localized and the modulation with age is found in the amplitude, connectivity, and panspectral burst probability. It is demonstrated that the wearable OPM-MEG could be used in children as a quite practical and easy-to-deploy neuroimaging method with performance as good as conventional MEG. With both good spatial (several millimeters) and temporal (milliseconds) resolution, it provides a novel and powerful technology for neurodevelopment research and clinical applications not limited to somatosensory areas.

We thank the reviewer for their summary, and their time in reviewing our manuscript.

The conclusions of this paper are mostly well supported by data acquired under the proper method. However, some aspects of data analysis need to be improved and extended.1. The colour bars selected for the pseudo-T-static pictures of beta modulation in Figures 2 and 3, which are blue/black and red/black, are not easily distinguished from the anatomical images which are grey-scale. A colour bar without black/white would make these figures better. The peak point locations are also suggested to be marked in Figure 2 and averaged locations in Figure 3 with an error bar.

Thank you for this comment which we certainly agree with. The colour scheme used has now been changed to avoid black. We have also added peak locations.

2. The data points in plots are not constant across figures. In Figures 3 and 5, they are classified into triangles and circles for children and adults, but all are circles in Figures 4 and 6.

Thank you! We apologise for the confusion. Data points are now consistent across plots.

3. Although MEG is much less susceptible to conductivity inhomogeneity of the head than EEG, the forward modulating may still be impacted by the small head profile. Add more information about source localization accuracy and stability across ages or head size.

This is an excellent point. We have added to our discussion relating to the accuracy of the forward model.

“…We failed to see a significant difference in the spatial location of the cortical representations of the index and little finger; there are three potential reasons for this. First, the system was not designed to look for such a difference – sensors were sparsely distributed to achieve whole head coverage (rather than packed over sensory cortex to achieve the best spatial resolution in one area (22)). Second, our “pseudo-MRI” approach to head modelling (see Methods) is less accurate than acquisition of participantspecific MRIs, and so may mask subtle spatial differences. Third, we used a relatively straightforward technique for modelling magnetic fields generated by the brain (a single shell forward model). Although MEG is much less susceptible to conductivity inhomogeneity of the head than EEG, the forward model may still be impacted by the small head profile. This may diminish spatial resolution and future studies might look to implement more complex models based on e.g. finite element modelling (23). Finally, previous work (24) suggested that, for a motor paradigm in adults, only the beta rebound, and not the power reduction during stimulation, mapped motortopically. This may also be the case for purely sensory stimulation. Nevertheless, it remains the case that by placing sensors closer to the scalp, OPM-MEG should offer improved spatial resolution in children and adults; this should be the topic of future work…”

**Recommendations for the authors:**

**Reviewer #2 (Recommendations For The Authors):**
Major items to further test include the differing number of trials, the windowing issue, and the focus on motor findings in the intro and discussion. First, I would recommend the authors adjust the number of trials in adults to equate them between groups; this will make their developmental effects easier to interpret.

Thank you for raising this important point. This has now been done and appears in our supplementary information as discussed above.

Second, to discern which responses are exhibiting developmental effects, the authors need to contrast the 0.3-0.8 window with the later window (2.5-3.0), not the window that appears to have the PMBR-like response. This artificially accentuates the response. I also think they should image the 1.0-1.5 vs 2.5-3.0s window to determine whether the response in this time window is in the same location as the decrease and then contrast this for beta differences.

We completely understand this point, which relates to separating the reduction in beta amplitude during stimulation and the rebound post stimulation. However, as explained above, doing so unambiguously would require the use of much longer trials. Here we were only able to measure stimulus induced beta modulation (distinct from the separate contributions of the task induced beta power reduction and rebound). It may be that future studies, with >10 s trial length, could probe the role of the PMBR, but such studies require long paradigms which are challenging to implement with children.

Third, changing the framing of the study to highlight the somatosensory developmental literature would also be an improvement.

We have added to our introduction a stated in the responses above.

Finally, the connectivity analysis on data from a somatosensory task did not make sense given the focus of the study and should be removed in my opinion. It is very difficult to interpret given past studies used resting state data and one would expect the networks to dynamically change during different parts of the current task (i.e., stimulation versus baseline).

We appreciate the point regarding connectivity. However, it was our intention to examine the developmental trajectory of beta oscillations, and a major role of beta oscillations is in mediating connectivity. It is true that most studies are conducted in the resting state (or more recently – particularly in children – during movie watching). The fact that we had a sensory task running is a confound; nevertheless, the connectivity we derived in adults bears a marked similarity to that from previous papers (e.g. (25)) and we do see significant changes with age. We therefore believe this to be an important addition to the paper and we would prefer to keep it.

References

(1) Holmes, N., Bowtell, R., Brookes, M. J. & Taulu, S. An Iterative Implementation of the Signal Space Separation Method for Magnetoencephalography Systems with Low Channel Counts.

Sensors 23, 6537 (2023).

(2) Boto, E. et al. Moving magnetoencephalography towards real-world applications with a wearable system. Nature (2018) doi:10.1038/nature26147.

(3) Holmes, M. et al. A bi-planar coil system for nulling background magnetic fields in scalp mounted magnetoencephalography. NeuroImage 181, 760–774 (2018).

(4) Seymour, R. A. et al. Using OPMs to measure neural activity in standing, mobile participants. NeuroImage 244, 118604 (2021).

(5) Rea, M. et al. A 90-channel triaxial magnetoencephalography system using optically pumped magnetometers. annals of the new york academy of sciences 1517, https://doi.org/10.1111/nyas.14890 (2022).

(6) Holmes, N. et al. Enabling ambulatory movement in wearable magnetoencephalography with matrix coil active magnetic shielding. NeuroImage 274, 120157 (2023).

(7) Pakenham, D. O. et al. Post-stimulus beta responses are modulated by task duration. NeuroImage 206, 116288 (2020).

(8) Fry, A. et al. Modulation of post-movement beta rebound by contraction force and rate of force development. Human Brain Mapping 37, 2493–2511 (2016).

(9) Pfurtscheller, G. & Lopes da Silva, F. H. Event-related EEG/MEG synchronization and desynchronization: Basic principles. Clin Neurophysio 110, 1842–1857 (1999).

(10) Seedat, Z. A. et al. The role of transient spectral ‘bursts’ in functional connectivity: A magnetoencephalography study. NeuroImage 209, 116537 (2020).

(11) Baker, A. P. et al. Fast transient networks in spontaneous human brain activity. eLife 2014, 1867 (2014).

(12) Vidaurre, D. et al. Spectrally resolved fast transient brain states in electrophysiological data. NeuroImage 126, 81–95 (2016).

(13) Gaetz, W. & Cheyne, D. Localization of sensorimotor cortical rhythms induced by tactile stimulation using spatially filtered MEG. NeuroImage 30, 899–908 (2006).

(14) Cheyne, D. et al. Neuromagnetic imaging of cortical oscillations accompanying tactile stimulation. Cognitive Brain Research 17, 599–611 (2003).

(15) van Ede, F., Jensen, O. & Maris, E. Tactile expectation modulates pre-stimulus β-band oscillations in human sensorimotor cortex. NeuroImage 51, 867–876 (2010).

(16) Salenius, S., Schnitzler, A., Salmelin, R., Jousmäki, V. & Hari, R. Modulation of Human Cortical Rolandic Rhythms during Natural Sensorimotor Tasks. NeuroImage 5, 221–228 (1997).

(17) Cheyne, D. O. MEG studies of sensorimotor rhythms: A review. Experimental Neurology 245, 27–39 (2013).

(18) Kilavik, B. E., Zaepffel, M., Brovelli, A., MacKay, W. A. & Riehle, A. The ups and downs of beta oscillations in sensorimotor cortex. Experimental Neurology 245, 15–26 (2013).

(19) Bauer, M., Oostenveld, R., Peeters, M. & Fries, P. Tactile Spatial Attention Enhances Gamma-Band Activity in Somatosensory Cortex and Reduces Low-Frequency Activity in Parieto-Occipital Areas. J. Neurosci. 26, 490–501 (2006).

(20) Barone, J. & Rossiter, H. E. Understanding the Role of Sensorimotor Beta Oscillations. Frontiers in Systems Neuroscience 15, (2021).

(21) Rayson, H. et al. Bursting with Potential: How Sensorimotor Beta Bursts Develop from Infancy to Adulthood. J Neurosci 43, 8487–8503 (2023).

(22) Hill, R. M. et al. Optimising the Sensitivity of Optically-Pumped Magnetometer Magnetoencephalography to Gamma Band Electrophysiological Activity. Imaging Neuroscience (2024) doi:10.1162/imag_a_00112.

(23) Stenroos, M., Hunold, A. & Haueisen, J. Comparison of three-shell and simplified volume conductor models in magnetoencephalography. NeuroImage 94, 337–348 (2014).

(24) Barratt, E. L., Francis, S. T., Morris, P. G. & Brookes, M. J. Mapping the topological organisation of beta oscillations in motor cortex using MEG. NeuroImage 181, 831–844 (2018).

(25) Rier, L. et al. Test-Retest Reliability of the Human Connectome: An OPM-MEG study. Imaging Neuroscience (2023) doi:10.1162/imag_a_00020.